# The ice nucleating activity of Arctic sea surface microlayer samples and marine algal cultures

Luisa Ickes[1,2*], Grace C. E. Porter[3], Robert Wagner[2], Michael P. Adams[3], Sascha Bierbauer[2], Allan K. Bertram[4], Merete Bilde[5], Sigurd Christiansen[5], Annica M. L. Ekman[1], Elena Gorokhova[6], Kristina Höhler[2], Alexei A. Kiselev[2], Caroline Leck[1], Ottmar Möhler[2], Benjamin J. Murray[3], Thea Schiebel[2], Romy Ullrich[2], and Matthew Salter[6]

[1]Department of Meteorology & Bolin Centre for Climate Studies, Stockholm University, Stockholm, Sweden
[2]Institute of Meteorology and Climate Research, Karlsruhe Institute of Technology, Karlsruhe, Germany
[3]School of Earth and Environment, University of Leeds, Leeds, United Kingdom
[4]Department of Chemistry, University of British Columbia, Vancouver, Canada
[5]Department of Chemistry, Aarhus University, Aarhus, Denmark
[6]Department of Environmental Science and Analytical Chemistry & Bolin Centre for Climate Studies, Stockholm University, Stockholm, Sweden
[*]Now at: Department of Space, Earth and Environment, Chalmers, Gothenburg, Sweden

**Correspondence:** Luisa Ickes (luisa.ickes@misu.su.se)

**Abstract.** In recent years, sea spray and the biological material it contains has received increased attention as a source of ice nucleating particles (INPs). Such INPs may play a role in remote marine regions, where other sources of INPs are scarce or absent. In the Arctic, these INPs can influence water-ice partitioning in low-level clouds and thereby the cloud lifetime, with consequences for the surface energy budget, sea ice formation and melt, and climate. Marine aerosol is of diverse nature, so identifying sources of INPs is challenging. One fraction of marine bioaerosol, phytoplankton and their exudates, has been a particular focus of marine INP research. In our study we attempt to address three main questions. Firstly, we compare the ice nucleating ability of two common phytoplankton species with Arctic seawater microlayer samples using the same instrumentation to see if these phytoplankton species produce ice nucleating material with sufficient activity to account for the ice nucleation observed in Arctic microlayer samples. We present first measurements of the ice nucleating ability of two predominant phytoplankton species, *Melosira arctica*, a common Arctic diatom species and *Skeletonema marinoi*, a ubiquitous diatom species across oceans worldwide. To determine the potential effect of nutrient conditions and characteristics of the algal culture, such as the amount of organic carbon associated with algal cells, on the ice nucleation activity, the *Skeletonema marinoi* was grown under different nutrient regimes. From comparison of the ice nucleation data of the algal cultures to those obtained from a range of sea surface microlayer (SML) samples obtained during three different field expeditions to the Arctic (ACCACIA, NETCARE, ASCOS) we found that although these diatoms do produce ice nucleating material, they were not as ice active as the investigated microlayer samples. Secondly, to improve our understanding of local Arctic marine sources as atmospheric INP we applied two aerosolisation techniques to analyse the ice nucleating ability of aerosolised microlayer and algae samples. The aerosols were generated either by direct nebulisation of the undiluted bulk solutions, or by the addition of the samples to a sea spray simulation chamber filled with artificial seawater. The latter method generates aerosol particles

using a plunging jet to mimic the process of oceanic wave-breaking. We observed that the aerosols produced using this approach can be ice active indicating that the ice nucleating material in seawater can indeed transfer to the aerosol phase. Thirdly, we attempted to measure ice nucleation activity across the entire temperature range relevant for mixed-phase clouds using a suite of ice nucleation measurement techniques- an expansion cloud chamber, a continuous flow diffusion chamber, and a cold stage. In order to compare the measurements made using the different instruments, we have normalised the data in relation to the mass of salt present in the nascent sea spray aerosol. At temperatures above 248 K some of the SML samples were very effective at nucleating ice, but there was substantial variability between the different samples. In contrast, there was much less variability between samples below 248 K. We discuss our results in the context of aerosol-cloud interactions in the Arctic with a focus on furthering our understanding of which INP types may be important in the Arctic atmosphere.

## 1 Introduction

Clouds have a strong impact on the energy balance and therefore play an important role in the Earth's climate system (Chahine, 1992; Boucher et al., 2013). They are particularly important in the high-latitudes, one of the regions most sensitive to global warming (Stocker et al., 2013), where they not only influence the energy budget (Garrett et al., 2009; Morrison et al., 2012), but also the subsequent melting and freezing of sea ice (Intrieri et al., 2002; Pithan and Mauritsen, 2014). As such, they are involved in several climate feedback processes. The radiative characteristics of clouds depend on their microphysical structure, e.g. if the cloud consists of water droplets or ice crystals. Mixed-phase clouds which are comprised of both ice crystals and super-cooled water droplets are common in the high Arctic (Shupe et al., 2006). Formation of liquid cloud droplets requires the presence of an aerosol particle that facilitates water vapour condensation on its surface (so-called cloud condensation nuclei – CCN). Aerosol particles are also necessary for the initiation of primary ice formation within these clouds by a process known as heterogeneous freezing (so-called ice nucleating particles – INP). Typically, only a small fraction of aerosol particles has the ability to nucleate ice. Despite increasing interest in INP (Szyrmer and Zawadzki, 1997; Hoose and Möhler, 2012; DeMott et al., 2010), it is still uncertain which types of aerosol particles that constitute good INP in the atmosphere (Kanji et al., 2017). Aerosol particles known to nucleate ice crystals by heterogeneous freezing in mixed-phase clouds include mineral dust, volcanic ash and primary biological particles, such as pollen, fungi and bacteria, and fragments of those (Hoose and Möhler, 2012). Those are aerosol particles with a predominantly terrestrial source. However, there are regions which are relatively isolated from terrestrial sources, such as the summer high Arctic, remote parts of North Atlantic, North Pacific and Southern Ocean. In such regions, sea spray aerosol could be an important source of INP (Burrows et al., 2013; Yun and Penner, 2013; Vergara-Temprado et al., 2017; Huang et al., 2018; McCluskey et al., 2018b, a; Creamean et al., 2019).

The potential for marine environments to act as sources of INPs was first investigated during the 1960s (see Table 1). This area of research has attracted renewed attention in more recent years. Indeed, recent observations indicate that biogenic material present at both the interface between the ocean and atmosphere, the so-called sea surface microlayer (SML), and within nascent sea spray aerosol can be ice active, e.g. Knopf et al. (2011); Wilson et al. (2015); DeMott et al. (2016); Irish et al. (2017); Gong et al. (2020). Previous studies can be separated into three main groups: (i) ambient ice nucleation measurements in marine

environments, (ii) studies investigating the ice nucleating potential of seawater and SML samples, and (iii) studies concerned with the ice nucleating potential of different phytoplankton species and their exudates (Table 1). One of the key recent studies

concerned with whether sea spray aerosol contains significant amounts of INPs was conducted by DeMott et al. (2016) who examined the ice nucleation potential of laboratory generated nascent sea spray aerosol particles and compared their findings with measurements of ambient marine aerosol. Critically, they observed that laboratory generated sea spray aerosol has a similar ice nucleation activity to ambient marine aerosols and that the ice nucleating activity of nascent sea spray aerosol strongly increased in association with phytoplankton blooms. Given these observations, the authors conclude that the INP

present in sea spray aerosol are likely linked to organic matter associated with phytoplankton blooms. DeMott et al. (2016) also showed that different INP types were active at different temperatures. Despite the finding that significant amounts of ice active material are present in nascent sea spray aerosol, the measured number concentration of INP in ambient marine aerosol was still several orders of magnitude lower than equivalent measurements in ambient terrestrial aerosol. Another relevant study was conducted by Wilson et al. (2015) who analysed SML samples collected in the Atlantic and Arctic oceans. The ice

activity of these samples was highly variable with the temperature at which half of the sample droplets froze, the so-called median freezing temperature, ranging from approximately 265 to 248 K. Based on tests with samples that have been filtered and heated, these authors concluded that submicron biogenic material was likely responsible for the ice activity of seawater samples from a range of locations. This suggests that whole cells are not responsible for the observed ice nucleation (Schnell and Vali, 1975; Wilson et al., 2015; Irish et al., 2017). Further, exudates of the marine diatom *Thalassiosira pseudonana*, a

widespread phytoplankton species, have been shown to nucleate ice (Knopf et al., 2011; Wilson et al., 2015; Ladino et al., 2016); hence it has been proposed that organic material associated with phytoplankton cell exudates may explain the ice nucleation activity of marine SML samples. However, Knopf et al. (2011) also found that intact cells are effective INP in the mixed-phase temperature regime. Another hypothesis is that bacteria play a role as shown by e.g. Fall and Schnell (1985).

Motivated by these previous studies we have analysed the freezing potential of two common phytoplankton species, *Melosira*

*arctica* and *Skeletonema marinoi*. *Skeletonema marinoi* is a very common diatom species, especially in temperate coastal regions during the spring bloom (Kooistra et al., 2008). *Melosira arctica* on the other hand is the most productive algae in the Arctic Ocean (Booth and Horner, 1997). *Melosira arctica* was found along with polymer gels in high Arctic cloud water samples (Orellana et al., 2011). Environmental factors, such as light and nutrient supply, have a high potential to affect the biochemical composition of phytoplankton and thus biogenic exudate material. The degree to which both the flux and

composition of sea spray aerosol is affected by biological activity in the surface ocean is a longstanding question in the field. However, studies have suggested that the aerosol flux may not only be impacted by the absolute cell concentrations of phytoplankton but also their growing conditions, e.g. Alpert et al. (2015). Thus those environmental factors have an effect on the presence of INPs coming from marine sources as well. Therefore, algae grown under different nutrient regimes may differ in their INP ability, which is investigated in this study. *Skeletonema marinoi* was cultivated with different nutrition levels in

order to mimic nutrient limitation and growth inhibition in phytoplankton. This leads to a variation in the carbon content of each cell and thus in the cell suspensions, which enables us to investigate the resulting effects of different growth rates and cell

carbon content on ice nucleation. Our aim here was to investigate whether changing these cell properties has any impact on the ice nucleation activity of the phytoplankton.

Another goal of this study was to improve our understanding of whether Arctic marine regions may have local sources of marine INPs. Although it has been found that organic matter with marine origin is prevalent in aerosol particles present in the high Arctic during summer e.g. Leck et al. (2002) and that marine organic matter nucleates ice e.g. Wilson et al. (2015), the ice nucleating potential of the aerosolised organic matter has not been examined in detail for the Arctic region. Therefore, we have determined the heterogeneous ice nucleating ability of artificial seawater containing two phytoplankton species cultured in the laboratory along with samples of SML collected during a series of field campaigns in the North Atlantic and Arctic Oceans. Within this study, two different aerosolisation techniques were utilised to test the impact of the aerosol generation method on the ice nucleation behaviour of the resulting particles.

Measurements have been made with a variety of ice nucleation measurement techniques and all measurements were conducted under conditions relevant for mixed-phase clouds, i.e. above about 235 K and at water saturation. We have utilised a number of different experimental methods to derive the ice nucleating ability of our samples, with the ultimate goal of merging these different measurements across the full temperature range relevant for mixed-phase clouds. Through comparison of the ice nucleation activity of artificial seawater containing *Melosira arctica* with that of the SML samples we aim to shed light on how representative relevant algal cultures are for Arctic marine INP.

A description of the methods of sample collection and cultivation as well as the experimental setup and ice nucleation measurement techniques are introduced in Sect. 2. The results of the ice nucleation measurements and a comparison with previous marine INP measurements found in the literature are presented in Sect. 3. Since we have made measurements across the full temperature range relevant for mixed-phase clouds (273.15 K until 233.15 K) this section is split into three parts. The first part (Sect. 3.1) focuses on the measurements at temperatures above 248 K referred to as the "high temperature regime" throughout this article, while the second part (Sect. 3.2) focuses on the measurements conducted at temperatures below 248 K referred to as the "low temperature regime" throughout this article. In the final part (Sect. 3.3) we present an integrated spectrum over the full temperature range. Finally, we conclude this study with a summary of the major findings and discussion of potential atmospheric implications of our results (Sect. 4).

## 2 Methods and experimental setup

To determine the ice nucleating ability of our samples we have used three independent methods (Fig. 1). Firstly, bulk cell suspensions of the algal cultures and field samples were aerosolised using a nebuliser and the generated particles were injected into the Aerosol Interaction and Dynamics in the Atmosphere (AIDA) aerosol and cloud chamber (Möhler et al., 2008). The ice nucleation behaviour of the particles was then either measured in situ in the AIDA chamber by performing an expansion cooling experiment, or by probing the particles with a continuous flow diffusion chamber (CFDC) called INKA [Ice Nucleation instrument of the KArlsruhe Institute of Technology; Schiebel (2017)]. Secondly, for a subset of the samples, a certain volume of the bulk solutions was added to 20 L of artificial seawater in the mobile Aarhus University sea spray simulation chamber

**Table 1.** An overview of previous laboratory and field studies which have either investigated nascent sea spray aerosol particles as INP or ambient INP in marine regions (including SML/seawater samples) in the temperature range of mixed-phase clouds. The location ("Loc.") of each of the field studies is given. Laboratory studies are indicated as "Lab". The "Data" column indicates how the ice nucleation activity was estimated- usual measures are as amount of INP per $m^3$ or L, the frozen fraction $FF$ as a function of temperature or the median freezing temperature $T_{50}$, i.e. the temperature at which 50% of the droplets were frozen. Where relevant, the "Subst." column states specific substances or species that were studied. The column "Instr." provides information about the instrument(s) used in the study. Here different cloud chambers are simplified by the term "cloud chambers", with the reference of the basic principle given in the index (1: Warner (1957), 2: Bigg (1957), 3: Bigg et al. (1963), 4: Langer et al. (1967), 5: Stevenson (1968), 6: Gagin and Arroyo (1969), 7: Langer and Rodgers (1975)). CFDC stands for continuous flow diffusion chamber. Cold stage, freezing assays etc. are all described with the term "drop freez.".

(i) Ambient ice nucleation measurements in marine environments

| Study | Loc. | Data | Subst. | Instr. |
|---|---|---|---|---|
| Kline and Brier 1958 | Washington DC | INP/L | Airborne | Cloud chamber[1] |
| Isono et al. 1959 | Tokyo (Pacific) | INP/L | Airborne | Cloud chamber[2] |
| Battan and Riley 1960 | Arizona (Gulf of Mexico) | INP/L | Airborne | Cloud chamber[1] |
| Kline 1960 | Washington DC | INP/L | Airborne | Cloud chamber[1] |
| Bigg 1973 | Southern Ocean (SO) | INP/$m^3$ | Airborne | Filter & cloud chamber[6] |
| Radke et al. 1976 | Alaska | INP/L | Airborne | Filter & dyn. chamber[4] |
| Schnell 1977 | Canada (Atlantic) | INP/$m^3$ | Airborne | Filter & drop freez. |
| Flyger and Heidam 1978 | Northern Greenland | INP/L | Airborne | Filter & cloud chamber[5] |
| Borys and Grant 1983 | Arctic | INP/$m^3$ | Airborne | Filter & cloud chamber[5] |
| Nagamoto et al. 1984 | Florida | INP/$m^3$ | Airborne | Filter & cloud chamber[7] |
| Fountain and Ohtake 1985 | Alaska | INP/L | Airborne | Filter & cloud chamber[3] |
| Rosinski et al. 1986 | Pacific | Freez. T, INP/$m^3$ | Airborne | Filter & drop freez. & cloud chamber[7] |
| Rosinski et al. 1987 | Pacific | Freez. T, INP/$m^3$ | Airborne | Filter & dyn. chamber[7] |
| Borys 1989 | Arctic | INP/$m^3$ | Airborne | Filter & cloud chamber[7] |
| Bigg 1990 | SO, Hawaii | INP/$m^3$ | Airborne | Filter & cloud chamber[3,5,6] |
| Rosinski et al. 1995 | East China Sea | INP/$m^3$ | Airborne | Filter & dyn. chamber[7] |
| Rosinski 1995 | Washington State (North Pacific) | INP/$m^3$ | Airborne | Filter & dyn. chamber[7] |
| Bigg 1996 | Arctic | INP/$m^3$ | Airborne | Filter & cloud chamber[5] |
| DeMott et al. 2016 | Caribbean, Arctic, Canada, Pacific, Lab (MART) | INP/L | Airborne | CFDC, filter & drop freez. |
| Mason et al. 2015 | Canada (North Pacific) | INP/L | Airborne | Filter & drop freez. |
| Ladino et al. 2016 | Canada (North Pacific) | INP/L | Airborne | CFDC |
| Creamean et al. 2018 | Arctic | INP/L | Airborne | Filter & drop freez. |
| McCluskey et al. 2018 | Mace head | INP/L | Airborne | CFDC, filter & drop freez. |
| McCluskey et al. 2018 | SO | INP/$m^3$ | Airborne | CDFC, filter & drop freez. |
| Si et al. 2018 | Canada/Arctic | INP/L | Airborne | Filter & drop freez. |
| Welti et al. 2018 | Atlantic | INP/$m^3$ | Airborne | CFDC, filter & drop freez. |

(i) Ambient ice nucleation measurements in marine environments

| Study | Loc. | Data | Subst. | Instr. |
|---|---|---|---|---|
| Creamean et al. 2019 | Arctic | INP/L | Airborne | Filter & drop freez. |
| Gong et al. 2019 | Cyprus | INP/L | Airborne | Filter & drop freez. |
| Irish et al. 2019a | Arctic | INP/L | Airborne | Filter & drop freez. |
| Ladino et al. 2019 | Gulf of Mexico | INP/L | Airborne | Filter & drop freez. |
| Si et al. 2019 | Arctic | INP/L | Airborne | Filter & drop freez. |
| Wex et al. 2019 | Arctic | INP/L | Airborne | Filter & drop freez. |
| Hartmann et al. 2020 | Arctic | INP/L | Airborne | Filter & drop freez. |
| Gong et al. 2020 | Cape Verde (Atlantic) | INP/L | Airborne | Filter & drop freez. |
| Welti et al. 2020 | Arctic, Atlantic, Pacific, SO | INP/m$^3$ | Airborne | CFDC, filter & drop freez. |

called AEGOR (Christiansen et al., 2019). Aerosol particles generated by bubble bursting in AEGOR were injected into the AIDA chamber in the same manner as the particles generated using the nebuliser and their ice nucleation activity was measured both in AIDA expansion cooling experiment and with INKA. Thirdly, the INP abundance within the liquid samples used to generate aerosols was determined using the microliter nucleation by immersed particle instrument ($\mu$l-NIPI), where droplets of the bulk solutions were pipetted onto a cold stage (Whale et al., 2015).

Additionally, it was investigated if material from the same algal cultures and SML samples affects the ability of sea spray aerosols to act as CCN. The measurements of the CCN-derived hygroscopicity and the implication on Arctic clouds are presented in a companion study, see Christiansen et al. (2020, submitted to J. Geophys. Res.).

### 2.1 Samples and sample treatment

Two types of samples were investigated in this study: algal cultures (*Skeletonema marinoi* and *Melosira arctica*) and SML
samples. One diatom species (*Skeletonema marinoi*) was grown under different conditions. The SML samples were collected during three field expeditions in the Arctic region [ACCACIA (Wilson et al., 2015), NETCARE (Irish et al., 2019b) and ASCOS (Gao et al., 2012)]. Table 2 provides an overview of how the samples were analysed and summarises all the measurements conducted during this campaign.

### Culture conditions and nutrient regimes for algae

The two diatoms were cultured axenically in Guillard's f/2+Si medium in two-liter glass bottles on a shaking table (0.5 rpm/min) inside a climate chamber. Algal growth rate and number of cells per colony were monitored using the cell counter TC20 (Bio-Rad). *Skeletonema marinoi* (CCAP 1077/5; Göteborg University Marine Algal Culture Collection, GUMACC) was isolated from the Long Island Sound (Milford Harbour, USA). *Melosira arctica* (MATV-1402; Helsinki University) originated

(ii) Studies investigating the ice nucleating potential of seawater and SML samples

| Study | Loc. | Data | Subst. | Instr. |
|---|---|---|---|---|
| Birstein and Anderson 1953 | Artificial | Freez. T | Sea salt | Cloud chamber |
| Brier and Kline 1959 | Washington DC | INP/L | Seawater | Cloud chamber[1] |
| Schnell and Vali 1975 | Pacific (N/S), Caribbean Atlantic | INP/m$^3$ | Seawater | Drop freez. |
| Schnell and Vali 1976 | Canada, California, Bahamas | INP/m$^3$ | Seawater | Drop freez. |
| Schnell 1977 | Atlantic (Canada) | INP/m$^3$ | Seawater | Drop freez. |
| Parker et al. 1985 | Antarctica | *FF* | Sea ice | Drop freez. |
| Rosinski et al. 1988 | Gulf of Mexico | INP/m$^3$ | Seawater | Cloud chamber[7] |
| Wilson et al. 2015 | Arctic, Canada N. Pacific, Atlantic | *FF*, $n_m$ | Seawater | Drop freez. |
| Irish et al. 2017 | Arctic | *FF* | Seawater | Drop freez. |
| McCluskey et al. 2017 | Lab (MART) | INP/L | Seawater | CFDC, filter & drop freez. |
| Irish et al. 2019b | Arctic | INP/L, *FF* | Seawater | Drop freez. |
| Creamean et al. 2019 | Arctic | INP/L | Seawater | Drop freez. |
| Wilbourn et al. 2020 | North Atlantic | Freez. T, *FF* | Aerosolized Sea water | Drop freez. |
| Gong et al. 2020 | Cape Verde (Atlantic) | INP/L | Seawater | Drop freez. |
| Wolf et al. 2020 | Florida Straits, North Pacific | Active site density | Sea water | CFDC |

(iii) Studies concerned with the ice nucleating potential of different phytoplankton species and their exudates

| Study | Loc. | Data | Subst. | Instr. |
|---|---|---|---|---|
| Schnell 1975 | Lab | INP/m$^3$ | Phytoplankton | Drop freez. |
| Parker et al. 1985 | Lab | *FF* | Mar. bacteria | Drop freez. |
| Fall and Schnell 1985 | Lab | $T_{50}$ | Mar. bacteria | Drop freez. |
| Alpert et al. 2011 | Lab | Freez. T | Aqu. NaCl, diatoms | Drop freez. |
| Alpert et al. 2011 | Lab | Freez. T | Phytoplankton | Drop freez. |
| Knopf et al. 2011 | Lab | Freez. T | Mar. diatoms | Drop freez. |
| Ladino et al. 2016 | Lab | *FF* | Phytoplankton, mar. bacteria | CFDC |
| McCluskey et al. 2017 | Lab | INP/L | Phytoplankton | CFDC, filter & drop freez. |
| DeMott et al. 2018 | Lab | $T_{50}$ | Fatty acids | Drop freez. |
| Tesson and Šantl Temkiv 2018 | Lab | Freez. T | Micro-algae | Drop freez. |
| Wilbourn et al. 2020 | North Atlantic | Freez. T | Phytoplankton | Drop freez. |

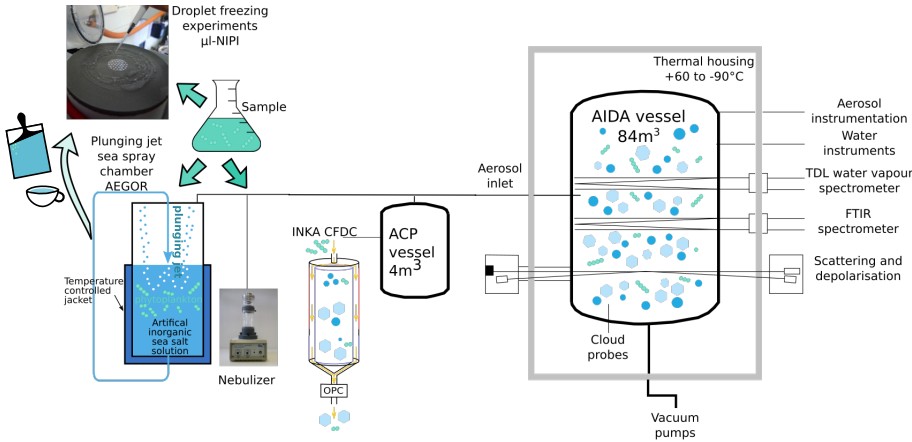

**Figure 1.** Schematic of the various aerosolisation (sea spray chamber AEGOR and nebuliser) and ice nucleation [Aerosol Interaction and Dynamics in the Atmosphere (AIDA) aerosol and cloud chamber, Ice Nucleation Instrument of the Karlsruhe Institute of Technology (INKA) and microliter nucleation by immersed particle instrument ($\mu$l-NIPI)] measurement techniques employed in this study.

from the Western Gulf of Finland, the Baltic Sea.

*Skeletonema marinoi* (SM) was grown at 26 PSU, 293.15 K using a 12:12 h light:dark cycle at $90\,\mu$mol photons m$^{-2}$ s$^{-1}$. Concentrations of nitrate and phosphate in the media were adjusted to conform to three experimental conditions in order to manipulate growth rates and cell carbon content: (1) nutrient-replete conditions (SM100; high growth, high nutrient content of cells), (2) 60% nutrient-saturation (SM60; high growth but low nutrient content), and (3) low-nutrient treatment (SM10; low growth, low nutrient content). The respective nitrate and phosphate concentrations were 5 and 1 $\mu$M in SM100, 3 and 0.6 $\mu$M in

SM60, and 0.2 and 0.1 $\mu$M in SM10 treatments. Nutrients, but also their stoichiometric ratios, determine growth. In treatment (1), we had algae that were both dividing fast and had a large cell size. In treatment (2), the cell division rate was high, but the cell size was small (phosphorus limitation). In treatment (3), both parameters were low (nitrogen and phosphorus limitation). The algae were harvested, i.e. the entire culture volume was transferred to a plastic bag and frozen, when reaching a density of $\sim 3 \times 10^5$ and $\sim 5 \times 10^6$ cells/mL in the nutrient-replete and nutrient-sufficient (SM100 and SM60) conditions, respectively.

Due to poor growth in SM10, the culture was harvested simultaneously with the other two treatments before reaching comparable cell densities.

*Melosira arctica* (MA) was grown at 6 PSU, 278.15 K and a 16:8 h light:dark cycle at $60\,\mu$mol photons m$^{-2}$ s$^{-1}$ and harvested when they reached $\sim 2 \times 10^5$ cells/mL. This culture is referred to as MA100.

Immediately after collection, the harvested algae were frozen for storage and transport at 193.15 K. We assume that freezing

the samples does not influence the results of the experiments, an assumption supported by the literature (Schnell and Vali, 1976; Irish et al., 2019b). Prior to freezing, a sub-sample of known volume from each species/treatment was collected on a 0.2 $\mu$m filter for dry weight (DW), C and N analysis. The non-purgeable organic carbon content and the water activity of each sample was measured after the experiments. These values are summarised in Table 3.

**Table 2.** Overview of the measurements conducted in this study. The first column lists all the different samples investigated (see Section 2.1) including information on the campaigns during which the field samples were collected. The type of the sample is given in the second column. The aerosolisation techniques used for the AIDA measurements is denoted in the third column while the fourth column lists all the ice nucleation instruments used to probe the sample. The fourth column shows the date of the experiments. The fifth and six column presents the results of the fitted aerosol size distribution: the median particle diameter $D$ and the width (geometric standard deviation) for all AIDA expansions (rounded to two digits after the comma).

| Sample name | Type | Aerosolisation techniques (AIDA) | Instruments | Date (AIDA expansion) | Fitted med. $D$ [$\mu$m] (AIDA) | Fitted std. dev. (AIDA) |
|---|---|---|---|---|---|---|
| Sigma-Aldrich sea salt | Artificial | Nebuliser, AEGOR | AIDA, $\mu$l-NIPI | 27.01.2017, 30.01.2017, 06.02.2017 | 0.59 0.31 0.43 | 1.41 2.34 2.68 |
| SM100 | Cultured | Nebuliser, AEGOR | AIDA, $\mu$l-NIPI, INKA | 06.02.2017, 21.02.2017, 07.02.2017, 08.02.2017 | 0.71 0.76 0.43 0.46 | 1.38 1.37 2.46 2.23 |
| SM60 | Cultured | Nebuliser | AIDA, $\mu$l-NIPI, INKA | 08.02.2017 | 0.73 | 1.43 |
| SM10 | Cultured | Nebuliser, AEGOR | AIDA, $\mu$l-NIPI, INKA | 16.02.2017, 17.02.2017 | 0.72 0.49 | 1.47 2.83 |
| MA100 | Cultured | Nebuliser, AEGOR | AIDA, $\mu$l-NIPI, INKA | 22.02.2017, 23.02.2017 | 0.41 0.74 | 1.33 2.8 |
| STN2 (NETCARE) | SML | Nebuliser | AIDA, $\mu$l-NIPI | 10.02.2017 | 0.82 | 1.30 |
| STN3 (NETCARE) | SML | Nebuliser | AIDA, $\mu$l-NIPI, INKA | 15.02.2017 | 0.77 | 1.38 |
| STN7 (NETCARE) | SML | Nebuliser | AIDA, $\mu$l-NIPI, INKA | 15.02.2017 | 0.77 | 1.39 |
| SML5 (ACCACIA) | SML | Nebuliser, AEGOR | AIDA, $\mu$l-NIPI, INKA | 01.02.2017, 02.02.2017 | 0.59 0.7 | 1.47 2.61 |
| SML8 (ACCACIA) | SML | Nebuliser, AEGOR | AIDA, $\mu$l-NIPI, INKA | 31.01.2017 | 0.88 0.40 | 1.21 2.9 |
| SML16 (ACCACIA) | SML | Nebuliser | AIDA, $\mu$l-NIPI, INKA | 03.02.2017 | 0.86 | 1.25 |
| SML17 (ACCACIA) | SML | Nebuliser | AIDA, $\mu$l-NIPI, INKA | 09.02.2017 | 0.83 | 1.36 |
| SML19 (ACCACIA) | SML | Nebuliser | AIDA, $\mu$l-NIPI, INKA | 03.02.2017 | 0.90 | 1.27 |
| ASCOS (< 5 kDa) | SML | Nebuliser | AIDA, $\mu$l-NIPI | 23.02.2017 | 0.75 | 1.43 |
| ASCOS (foam) | SML | Nebuliser | AIDA, $\mu$l-NIPI | 24.02.2017 | 0.85 | 1.3 |
| ASCOS (5 kDa to 0.22 $\mu$m) | SML | Nebuliser | AIDA, $\mu$l-NIPI | 24.02.2017 | 0.18 | 1.27 |

**Table 3.** Characteristics of the bulk samples used during the study: non-purgeable organic carbon content, the water activity of the artificial seawater, algal cultures and two SML samples, the algae cells per mL of the cultures and the carbon cell content of the cultures. For the diluted samples we give in parentheses how many mL of sample where added to 20 L of artificial seawater (3.5 wt% solution of the synthetic Sigma-Aldrich sea salt mixture in ultrapure water) in the AEGOR sea spray tank (see Sect. 2.2). For the samples indicated with "pure" the undiluted sample was used. The water activity of the samples was estimated directly using the dewpoint. These measurements were repeated three times, resulting in the standard deviations (STD) given here.

| Sample name | Non-purge able organic carbon [mg C L$^{-1}$] | Water activity (dewpoint) | Water activity STD (dewpoint) | Algae cells [mL$^{-1}$] | Carbon cell content [$\mu$gC mL$^{-1}$] |
|---|---|---|---|---|---|
| SM100 (pure) | 14.3 | 0.9871 | 0.0004 | 5280000 | 105.6 |
| SM10 (pure) | 5.1 | 0.9916 | 0.0005 | 350000 | 9.8 |
| MA100 (pure) | 10.9 | 0.9861 | 0.0006 | 188700 | 245.31 |
| Sigma-Aldrich sea salt (pure) | 1.1 | 0.9854 | 0.0004 | | |
| SM100 (79 mL in AEGOR) | 2.3 | 0.9861 | 0.0008 | 20774 | 0.42 |
| SM100 (406 mL in AEGOR) | 1.7 | 0.9838 | 0.0006 | 105051 | 2.1 |
| SM10 (approx. 900 mL in AEGOR) | 0.9 | 0.9855 | 0.0002 | 15072 | 0.42 |
| MA100 (893 mL in AEGOR) | 3.2 | 0.9861 | 0.0006 | 1 | 10.49 |
| SML8 (200 mL in AEGOR) | 1.6 | 0.9866 | 0.0004 | | |
| SML5 (100 mL in AEGOR) | 1.1 | 0.9857 | 0.0002 | | |

**Field samples**

The SML samples were collected from different locations in the Arctic. A subset of the samples were collected during the Aerosol-Cloud Coupling and Climate Interactions in the Arctic (ACCACIA) expedition in July and August, 2013 in the Arctic Atlantic [East of Greenland and North of Spitsbergen, for more details see Wilson et al. (2015)]. Another subset of samples were collected as part of the Network on Climate and Aerosols: Addressing Key Uncertainties in Remote Canadian Environments (NETCARE) project during July and August, 2016 in the Eastern Canadian Arctic [for more details see Irish et al. (2019b)].

During ACCACIA and NETCARE a remote-controlled sampling catamaran was used for collection [ACCACIA: Knulst et al. (2003); Matrai et al. (2008); NETCARE: Shinki et al. (2012)]. Previous analysis of these samples in terms of ice nucleating ability can be found in the respective publications (Wilson et al., 2015; Irish et al., 2019b). The third subset of samples originates from the Arctic Summer Cloud Ocean Study (ASCOS) in August 2008 (Tjernström et al., 2014). The surface microlayer water was collected from an open lead using the same sampling catamaran used during the ACCACIA campaign. The sample

investigated in this study was collected on August 17 in 2008 at ca. 88°N and treated afterwards in three different ways. Two subsamples were subjected to a two-step ultrafiltration procedure. Firstly, the sample was passed through Millipore membrane

filters (nominal pore size 0.22 $\mu$m) under mild vacuum. Secondly, the filtered samples were ultrafiltered and diafiltered through a tangential flow filtration system (TFF, Millipore) equipped with cartridges with a molecular weight cut off of 5 kDa. The fraction that passed through the 0.22 $\mu$m filters but not the TFF system is referred to as high molecular weight dissolved organic matter (5 kDa to 0.22 $\mu$m). To obtain even greater separation into low molecular weight dissolved organic matter, sample which passed through the TFF system was further filtered in an Amicon®stirred cell ($< 5$ kDa). The third subsample is a foam layer sample. Seawater without pre-filtration was fed directly into a pre-cleaned glass tower (15.3 L, 2 m in height). Purified zero air was forced into the system through a sintered glass frit (nominal pore size 15 - 25 $\mu$m) from the bottom of the tower at a flow rate of 150 mL min$^{-1}$. After the bubble experiment, seawater at the uppermost layer (about 3 cm) together with foamy substances were slowly overflowed into a collecting flask by an additional feeding of seawater from the middle of the tower. The collected water from the top layer consisting of both foam and background seawater is referred to as foam layer sample. The foam sample should be similar to an unfiltered SML sample (as obtained during ACCACIA and NETCARE). More details on the methods of filtration applied during ASCOS can be found in Gao et al. (2012).

All samples were immediately frozen at 193.15 K for storage and transport. The field samples are labelled according to the original names in the respective publications: the samples originating from the field expedition ACCAIA are called SML (purple, green and turquoise colours in the figures), the samples from NETCARE STN (blue colours in the figures) and the samples from ASCOS are called ASCOS (red and yellow colours in the figures). The numbers refer to the original sample numbers.

## 2.2 Aerosolisation techniques

Two different techniques were used to aerosolise samples into the AIDA cloud chamber. Firstly, undiluted samples were aerosolised using an ultrasonic nebuliser (GA2400, SinapTec) and injected directly into the AIDA chamber. An injection period of 20-30 minutes was sufficient to fill the AIDA chamber with an aerosol number concentration of approx. 550 cm$^{-3}$. Secondly, we used the temperature-controlled sea spray simulation chamber, AEGOR, with the aim of generating bubble-bursting aerosols in a more representative manner (Christiansen et al., 2019). The sea spray tank was filled with 20 L of artificial seawater (3.5 wt% solution of the synthetic Sigma-Aldrich sea salt mixture, product number S9883, in ultrapure water). Sigma-Aldrich sea salt is nominally purely inorganic and should not contain any biological or other ice nucleating components. Thereafter, a certain volume of the investigated sample, as specified in Table 3, was added and the aerosol generation process was started. The cell concentrations of algae in the AEGOR tank (see Table 3) ranging from 1 to 10$^6$ cells mL$^{-1}$ are representative for a strong phytoplankton bloom (Henderson et al., 2008; Borkman and Smayda, 2009; Saravanan and Godhe, 2010; Suikkanen et al., 2011; Canesi and Rynearson, 2016). In AEGOR sea spray aerosols are generated by a plunging jet that entrains air into the sea spray tank and thus leads to bubble bursting, emitting aerosol particles to the head space (flow rate of the jet 5 L min$^{-1}$, nozzle diameter 4 mm). Bubble formation using this technique mimics bubble formation through wave breaking. Bubbles rising through the water column scavenge surface active organic material and transport it to the surface where it forms a microlayer. Subsequently, bubble-bursting transfers this surface active organic material to the aerosol phase.

Since the efficiency of particle generation by the sea spray simulation chamber was much lower than the nebuliser, injection of particles generated using this approach into the AIDA chamber was conducted over a period of 14-16 h, resulting in an aerosol particle concentration of approx. 300-400 $cm^{-3}$. Because of this time-consuming procedure, only a subset of the bulk solutions was used for aerosol generation with AEGOR (Table 2). The temperature of the AEGOR tank was set to 293.15 K for the SM culture samples, 277.15 K for the MA culture sample and 275.15 K for the SML samples.

Aerosolising an SML sample with a nebuliser is very different from aerosolisation due to bubble-bursting for a number of reasons. Firstly, only a small volume of sample is required for nebulisation so pure SML samples could be aerosolised (we had limited sample volume) while the sea spray simulation chamber requires a higher volume of sample as they were added to 20 L of artificial seawater (we used up to 900 mL sample volume). As such, the SML samples underwent significant dilution when added to artificial seawater in the sea spray simulation chamber. Secondly, the process of aerosol generation by bubble bursting
is quite different to aerosol generation in a nebuliser. As such, those aerosols generated in the sea spray simulation chamber are likely more representative of aerosols generated by oceanic bubble-bursting (Collins et al., 2014; King et al., 2012; Prather et al., 2013). Given these differences, comparison of the ice activity of aerosol generated by these two techniques should enable us to determine whether INP material is preferentially aerosolised by bubble-bursting.

We expect that aerosolisation of the samples with the nebulizer results in an upper estimate of INP because the undiluted
SML (or cultured) samples are aerosolised whereas AEGOR is aerosolising a dilution of the samples with artificial seawater, which could result in a lower estimate of INP. However, it is not only the dilution factor in the sea spray simulation chamber (see Table 3), which has to be accounted for. The aerosolisation process itself is different in AEGOR compared to the nebuliser. In the nebuliser the suspension is well mixed, while in AEGOR the aerosol particles are formed from an organic enriched surface microlayer at the top of the tank. That leads to different expectations depending on the sample type. For the SML samples
we would not expect such a huge difference due to this aspect. Here, we aerosolise in one case the pure well mixed SML (nebuliser), while in the other case we aerosolise the SML that has formed in AEGOR, which should be similar to the original SML sample. For the cultured samples, however, we would expect a larger influence. In AEGOR the phytoplankton material is floating at the surface of the tank leading to organic enriched aerosol particles during the aerosolisation, while the nebuliser might produce less enriched aerosol particles due to the mixing of the sample. Note that this might depend on the algae culture
as well. Another crucial aspect of the two different aerosolisation methods is the size distribution and the resulting chemical composition of the generated aerosol. It was demonstrated in the laboratory and as well measured in the field, that for sea spray aerosol the organic composition of the aerosol particles and the generated size distribution are related (O'Dowd, C. D. et al., 2004; Prather et al., 2013). One interesting aspect of our study is to see the influence of all the aspects mentioned above and to check if the diluted samples aerosolised with AEGOR show a similar or a lower freezing signal compared to the aerosolised
pure samples.

### 2.3 Aerosol size and number measurements

The aerosol particle number concentration was measured using a condensation particle counter (CPC3010, TSI). The aerosol particle number size distributions were measured with a scanning mobility particles sizer (SMPS, TSI; mobility diameter

0.014 - 0.820 $\mu$m) and an aerodynamic particle spectrometer (APS, TSI; aerodynamic diameter 0.523 - 19.81 $\mu$m). In the AIDA chamber, typically held at 250 K and a relative humidity of 78% during aerosol injection (see Sect. 2.4 for more details and an explanation on the low temperature), the aerosol particles were suspended as supercooled aqueous solution droplets. It is important to consider, however, that the size distribution measurements were done at room temperature (298 K) by sampling air from the cold interior of the aerosol chamber (Fig. 1). The water vapour content at 250 K corresponds to a relative humidity of only 2.4% after warming to 298 K (Murphy and Koop, 2005). We thus assume that the measured size distributions represent the effloresced, dry particle sizes of the algal culture and SML particles (Koop et al., 2000). A dynamic shape factor of 1.08 and a particle density of 2.017 g cm$^{-3}$ (Zieger et al., 2017) for sea salt were used to convert the mobility and aerodynamic diameters of the SMPS and APS measurements into the volume-equivalent spherical diameters. Fig. 2 shows the combined size spectra of the SMPS and APS measurements, plotted as surface area size distributions, for two exemplary aerosol particle populations produced by the nebuliser and AEGOR (SM100 and SML8).

The comparison of both aerosolisation techniques for the algae and the field samples shows that the nebuliser produces rather uniformly sized particles with a median diameter of about 0.8 $\mu$m in the surface area size distributions. In contrast, the bubble bursting process simulated in AEGOR leads to a much broader surface area size distribution with a smaller median diameter. However, both the nebulizer and AEGOR are not producing very narrow size distributions (see Fig. 2). The majority of our aerosolised samples yielded surface area size distributions very similar to those shown in Fig. 2. For each sample a log-normal fit was created based on least-squares. The fits are expressed as a function of the median equal-volume sphere diameter, the geometric standard deviation $\sigma$ and the aerosol surface area concentration. The median diameter of the particles generated with the nebuliser was typically in the range from 0.71 to 0.90 $\mu$m with a distribution width $\sigma$ between 1.21 and 1.47. Smaller particles with median diameters of 0.59, 0.41, and 0.18 $\mu$m were obtained for the SML5, MA100, and ASCOS (high mol. weight, 5 kDa - 0.22 $\mu$m) samples, respectively, which is probably related to lower salt concentrations in the respective solutions. Aerosol generation with AEGOR yielded median diameters between 0.4 and 0.7 $\mu$m and distribution widths $\sigma$ between 2.2 and 2.9.

## 2.4 Ice nucleation measurement techniques

The combination of instrumental methods used in this study facilitates measurement of the ice nucleating ability of marine organic aerosols over a wide temperature range. The ice nucleation activity was measured using three different ice nucleation instruments: AIDA, INKA, and the $\mu$l-NIPI, which all have their highest sensitivities in different temperature ranges. While the $\mu$l-NIPI is sensitive in the temperature regime above 248 K, AIDA and INKA are only sensitive in the temperature regime below 248 K for the type of samples analysed in this study. All three measurement techniques are explained in detail in the following sections.

**AIDA**

The AIDA facility comprises two aerosol chambers (Fig. 1) (Möhler et al., 2008). The term AIDA chamber refers to the 84.3 m$^3$ sized aluminium vessel that is enclosed in an isolating containment and can be operated at any temperature between

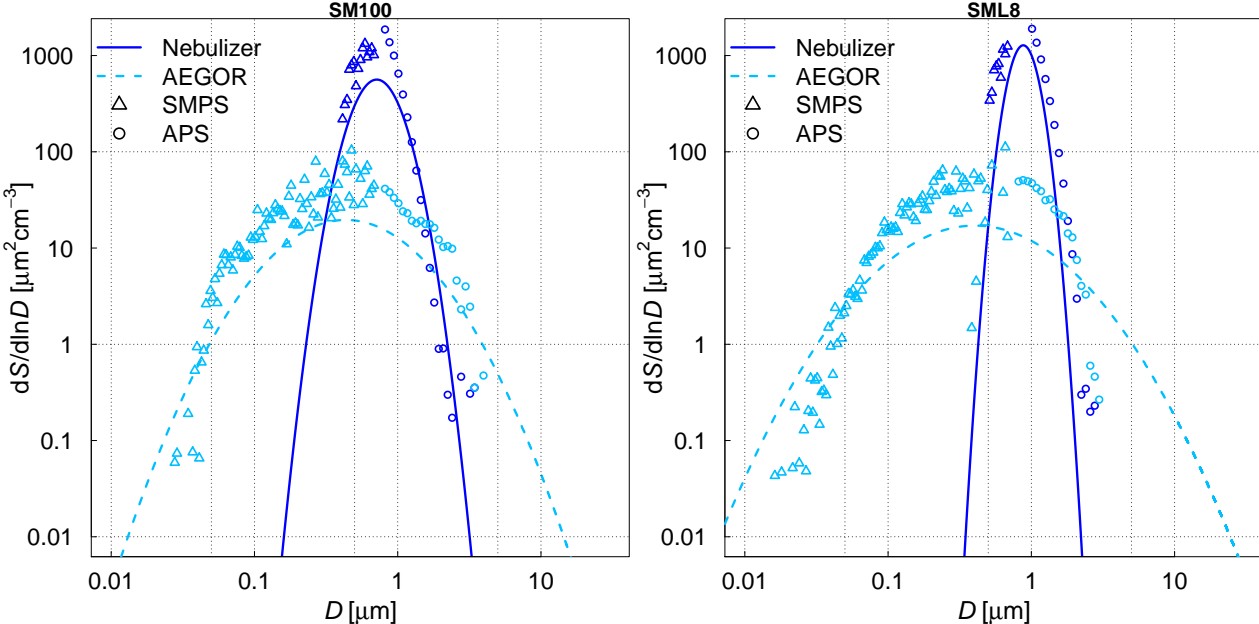

**Figure 2.** Measured size distributions and fits to the data for two different samples: an algae sample (SM100) and a field sample (SML8). The samples were aerosolised using a nebuliser (solid line) or the sea spray simulation chamber AEGOR (dashed line). The aerosol size measurements are done with an APS (circles) and a SMPS (triangles). $D$ denotes the equal-volume sphere diameter of the aerosol particles, $S$ the surface area concentration.

ambient and 183 K. A smaller 3.7 m³-sized stainless steel vessel is located in the vicinity of AIDA. It is referred to as the APC (aerosol preparation and characterisation) chamber and can only be operated at ambient temperature. As indicated in Sect. 2.2, the aerosol particles were directly injected into the AIDA chamber to probe their ice nucleation activity by expansion cooling

275 experiments. For practical reasons, the same aerosol particles were additionally injected into the APC chamber, acting as a reservoir for long-term measurements of the particles' ice nucleation behaviour with the INKA instrument (see next section) and for the CCN measurements (see Christiansen et al. 2020, submitted to J. Geophys. Res.).

The operation of the AIDA chamber as a cloud simulation chamber for studying ice nucleation has been thoroughly described previously (Möhler et al., 2003; Möhler et al., 2005; Wagner and Möhler, 2013). Briefly, a mechanical pump is used

280 for a controlled reduction of the chamber pressure starting from ambient to about 800 hPa. Expansion cooling generates supersaturations with respect to ice and/or supercooled liquid water, triggering the formation of ice crystals and supercooled water droplets by various nucleation mechanisms (Vali, 1985; Vali et al., 2015). In the present study, the ice nucleation activity of the algal cultures and SML samples was investigated in the immersion freezing mode at mixed-phase cloud temperatures. For aerosol injection, the AIDA chamber was typically held at a temperature of 250 K and a relative humidity with respect to

285 supercooled water ($RH_w$) of about 78%, as controlled by an ice layer on the inner walls of the aluminium vessel. $RH_w$ was measured in situ by tuneable diode laser (TDL) absorption spectroscopy with an uncertainty of $\pm 5\%$ (Fahey et al., 2014). With

increasing $RH_w$ during expansion cooling, the injected aqueous solution droplets continuously took up water vapour from the gas phase, and were finally activated to $\geq 10$ $\mu$m-sized cloud droplets when $RH_w$ exceeded 100%. The number concentration and size of the cloud droplets were measured with two optical particle counters (OPCs) Welas 1 and 2 (Palas GmbH) with an overall detection range of 0.7 - 240 $\mu$m. Cloud formation was typically observed after 3 K of expansion cooling, i.e., at a temperature of about 247 K. Whereas pure supercooled water droplets would only freeze homogeneously when the gas temperature further dropped to about 238 K during expansion cooling (Benz et al., 2005), the activated algal culture and SML aerosol particles exhibited heterogeneous ice nucleation modes due to immersion freezing at temperatures above 238 K. The number concentration of the nucleated ice crystals, $N_{ice}$, was separately deduced from the OPC records by using an optical threshold size to substract the scattering signals of the smaller-sized supercooled cloud droplets. By dividing $N_{ice}$ through the seed aerosol particle number concentration, the ice active fraction, $FF$, of the aerosol particle population was calculated. By further dividing $FF$ through the average dry surface area of a particle, $A_{aer}$ (determined from the size distribution measurements shown in Fig. 2) the ice nucleation active surface site density, $n_s$, of the polydisperse particle population could be computed, e.g. Hoose and Möhler (2012):

$$n_s(T) = \frac{FF(T)}{A_{aer}} \tag{1}$$

This equation is an approximation, which is valid for small values of $FF(T)$ (Hoose and Möhler, 2012) and was tested to be applicable for the dataset presented here. It is also assumed that $n_s$ is independent of size.

The uncertainty of the deduced ice nucleation active surface site densities ($n_s$) was estimated to $\pm 40\%$ (Ullrich et al., 2017). In the following we estimate a lower detection limit of $n_s$ in the AIDA experiments. The minimum detectable ice particle number concentration, as limited by the size of the detection volume of the OPC sensors, is about 0.05 cm$^{-3}$, equalling to one detected ice crystals in a sampling period of about 10 s. Together with the typical seed aerosol particle number concentration of about 500 cm$^{-3}$ (Sect. 2.2), the lower detection limit for $FF$ can thus be estimated to about $10^{-4}$. The average dry surface area of the aerosol particles generated with the nebuliser was around 1 $\mu$m$^2$, yielding a lower detection limit for $n_s$ of about $10^8$ m$^{-2}$ (Eq. 1). In comparison with recent literature $n_s$ values for laboratory and field sea spray aerosol particles (DeMott et al., 2016), $n_s$ only exceeded such values at temperatures below about 248 K. This illustrates why the starting temperature of the expansion cooling runs was chosen as low as 250 K, thus limiting the ice nucleation data to temperatures below about 247 K. During our study we also probed a number of samples (STN2, STN3 and SM100) at a higher starting temperature of 258 K. However, we did not observe any ice formation above the detection limit down to a temperature of 248 K. For this reason, the AIDA data cover the above-defined low temperature regime of the ice nucleation spectra.

In addition to the expansion cooling cycles with the algal and SML samples, we conducted three control runs with the synthetic Sigma-Aldrich sea salt mixture, both using AEGOR and the nebuliser for aerosol generation. Here, the deduced $n_s$ were close to the estimated detection limit of $1 \cdot 10^8$ m$^{-2}$ at temperatures between 247 and 238 K. The small amount of heterogeneously formed ice crystals could be due to traces of insoluble components in the synthetic salt mixture or due to ice nucleation on background aerosol particles in the cloud chamber. All aerosols exhibited $n_s$ values 2–50 times larger than this background signal (see Sect. 3.2). To account for possible contamination originating in the nebuliser or AEGOR a background

subtraction was conducted using these reference experiments with a pure Sigma-Aldrich sea salt solution and subsequent estimation of the average background $n_s$ value. The estimated background from these reference experiments was consistent and independent of temperature. It is higher for AEGOR compared to the nebuliser, probably due to the more complex setup of aerosolisation in the former.

## INKA

Most of the samples that were probed in the AIDA chamber were also tested on their ice nucleation activity using the INKA cylindrical continuous flow diffusion chamber (Schiebel, 2017). As explained above, the APC chamber was used as an aerosol particle reservoir for the INKA measurements. The APC chamber was held at 298 K and RH < 5%, meaning that the injected solution droplets generated with the nebuliser or AEGOR readily effloresced to form crystalline particles. Upon injection into the INKA instrument, aerosols are exposed to well controlled temperature and relative humidity conditions by flowing through a chamber with iced walls held at different temperatures. The sample air flow is sheathed by particle free synthetic air (initially dry) in order to position the aerosol lamina between the walls and to allow for the calculation of the thermodynamic conditions within the lamina (Rogers, 1988). The residence time of the aerosol is 10 to 15 s, depending on the actual settings. Any droplets that might have formed in this section will shrink in a subsequent chamber section with no temperature difference between the iced walls. The formed ice particles will persist in this so-called evaporation section. The thus increased size difference between droplets and ice particles at the chamber outlet allows for an easy ice particle detection with an optical particle counter (Climet CI-3100). INKA scans the ice nucleation activity by continuously increasing the sample's relative humidity at constant temperature settings. Due to a larger detection volume of the Climet OPC compared to the Welas sensors used in the AIDA experiments, the lower detection limit for $n_s$ with INKA is about $10^7$ m$^{-2}$. In inter-comparison studies using natural soil dust aerosol (DeMott et al., 2018a) or commercially available cellulose particles (Hiranuma et al., 2019) INKA has shown a good agreement with AIDA and other ice nucleation instruments. In the present study, most experiments have been conducted above 241.15 K to enable a clear differentiation from homogeneous freezing events and to allow direct comparison with AIDA results.

## $\mu$l-NIPI

The $\mu$l-NIPI is a cold stage instrument, used with a substrate to probe the ice nucleation in immersion mode of $\mu$l volume droplets (Whale et al., 2015). To do so, the droplets of the sample under investigation (if not explicitly otherwise mentioned this is a bulk sample) are pipetted onto a silanised glass slide, which serves as a hydrophobic substrate. It is a "bulk" technique analysing the suspension directly under the assumption that the sample is well mixed, so that particles are distributed randomly, and each droplet is representative of the sample as a whole, meaning each one has an approximately equal probability of containing an INP active at a given temperature. The droplets are then cooled at a rate of 1 K min$^{-1}$ until the droplets are all frozen. The temperature values of the individual freezing events are optically detected using a camera and offline analysis. The number of droplet freezing events detected throughout the temperature ramp are then converted into a fraction frozen at each temperature. This fraction frozen, or $FF$ curve, represents the raw freezing events. In order to calculate a concentration of INP

per liquid unit volume of sample, $K(T)$, the $FF$ must be thought of as the probability of freezing, and so the equation below can be used to deduce the cumulative nucleus concentration per unit volume of sample used (Vali, 1971):

$$FF(T) = \frac{N_{\text{frozen droplets}}(T)}{N_{\text{droplets}}} \tag{2}$$

$$K(T) = \frac{-\ln(1 - FF(T))}{V_{\text{droplet}}} \cdot D, \tag{3}$$

where $V_{\text{droplet}}$ is the volume of a droplet, $N_{\text{droplets}}$ is the total number of droplets on the cold stage at the beginning of the freezing experiment, $N_{\text{frozen droplets}}$ is the amount of droplets frozen at a certain temperature and $D$ is the dilution factor relative to the undiluted sample, relevant for the samples coming from AEGOR and a couple of dilution experiments conducted with the algal cultures (in all other cases $D$ is 1).

$K(T)$ can then be weighted to physical aspects of the sample such as the surface area of the particles or the mass of salt in the sample in order to directly compare to other instruments using the same sample.

In contrast to AIDA and INKA the $\mu$l-NIPI is sensitive to INP in a relatively high temperature range. Given the relatively large size of the pipetted droplets, this technique is better suited to the investigation of freezing by rare INPs i.e. there is a greater probability of having an INP within the droplet which subsequently freezes the whole droplet.

## 3 Results

In this section, we first address the ice nucleation measurements with the $\mu$l-NIPI instrument in the temperature regime above 248 K (Sect. 3.1). The AIDA and INKA results for temperatures below 248 K are presented in Sect. 3.2. Finally, Sect. 3.3 outlines an approach to combine the AIDA/INKA and $\mu$l-NIPI data into a single dataset to examine the ice nucleation behaviour of the algal cultures and Arctic SML samples over the full temperature range relevant for freezing in the mixed-phase cloud regime.

### 3.1 Temperature regime above 248 K (Bulk samples)

The frozen-fraction curves measured with $\mu$l-NIPI for the field and algal samples are shown in Fig. 3. Among the field samples there is a large spread in ice nucleation activity with a median freezing temperature $T_{50}$ ($FF$=0.5, i.e. half of the droplets are frozen) of approx. 262 to 245 K, i.e. a spread of 17 K. While the ice nucleation is very variable throughout the samples, the dependence on temperature (slope of the curves) is mostly similar. A number of the samples exhibited ice nucleation activity at relatively high temperatures (>263.15 K), with the ASCOS high molecular weight sample (ASCOS high mol. w., 5 kDa to 0.22 $\mu$m) and SML5 being the most ice active. Both algal samples studied were also ice active, although they were clearly less ice active than the field samples despite their relatively high cell concentration (compared to natural seawater). For example, the $T_{50}$ of the culture samples is approx. 252 to 246 K (range of 6 K), so within the colder part of the variability of the field samples (see Fig. 3). Further to this, no large differences (a difference of $T_{50}$ of approx. 5 K) were observed between the

different diatom species or when comparing the different nutrient conditions for SM. However, it should be noted that there was variability within the individual cultures and that storage changed the ice nucleation activity. For example, the same SM100 culture was delivered to AIDA in two separate bags. We refer to one bag as SM100a, the other one SM100b. A third sample (SM100c, a sub-sample of SM100b) was analysed two months after the campaign after having been stored at or below 253 K. SM100d, also a sub-sample of SM100b, was used for some further tests 10 months after the campaign (as well stored at or below 253 K). Note that the results of SM100d should be used with caution and not directly compared to the other ones, since this sample was unfrozen several times and stored for a quite a long period of time, which might not be ideal.

Comparing SM100a, SM100b and SM100c, it can be seen that the freezing properties of the SM100 sample is variable. SM100a and SM100b look similar, with most of the spectrum at low temperature in the background. They show differences at higher temperatures, where SM100a displayed activity but SM100b did not. SM100c showed different activity from SM100a and SM100b with the freezing shifted to higher temperatures that leads to the whole curve being outside of the background (compared to SM100b). The gradients between all three curves is also different, with SM100a having the shallowest slope. Additional to the variability of the sample itself (different bags - SM100a and SM100b), it seems that the sample changed with time, so age, storage and multiple freezing cycles may all have had effects on the sample.

The STN samples have been analysed previously using a similar droplet freezing technique albeit using a 10 times faster cooling rate (10 K min$^{-1}$) (Irish et al., 2019b). Comparison of these measurements with our measurements of the same samples highlight the differences. We observed up to an order of magnitude higher $K(T)$ values [and up to a 10 K difference for the same $K(T)$] than those reported in (Irish et al., 2019b), which might have been influenced by the difference in the cooling rate. The temperature at which 50% of the droplets are frozen has been shown to decrease with increased cooling rate in Wright and Petters (2013); Herbert et al. (2014), also this dependence was shown to be rather small. Nevertheless, a shift of 10 K for a factor of 10 change in cooling rate is unlikely. The SML samples from Wilson et al. (2015) were analysed using the same droplet freezing technique as in this study. Samples SML5, SML8 and SML16 exhibited ice activity at similar temperatures to those presented in Wilson et al. (2015), while samples SML17 and SML19 exhibited lower ice activity, with lower temperatures of freezing for the same fraction frozen. Therefore, we conclude that some samples were unaffected by long-term storage (being frozen at 193.15 K), while the activities of other samples changed. This indicates that some ice active components are altered through the freezing, storage and thawing process. Note that this contradicts earlier assumptions based on findings of Schnell and Vali (1976); Irish et al. (2019b). However, Polen et al. (2016) has shown that the biological INP Snomax is sensitive to storage. An alteration of INP characteristics of our microlayer samples indicates that they contain different ice active components which have different properties and may be related to different biological processes. In this paper we use the re-measured droplet freezing results to compare the ice nucleation activity between instruments.

The influence of bubbling the samples in the sea spray chamber AEGOR on the ice nucleation activity was investigated by comparing pure samples with three different sub-samples taken out of AEGOR after bubbling: one bulk sub-sample (collected from the bottom of AEGOR), one scoop sub-sample (collected by scooping a falcon tube along the surface liquid) and a microlayer sub-sample [collected by the glass-plate technique as per the methods of (Harvey, 1966)]. Note that all these samples are bulk samples. Upon introduction to AEGOR there was a significant dilution of the sample with artificial seawater

(Table 3). The ice nucleation activity of the SML5 sub-samples as described above is shown in Fig. 4. To take the dilution into account the data is plotted with respect to the volume of sample used, as INP/L (see Eq. 3). Interestingly, the bulk and microlayer sub-samples exhibit lower ice activity than the scoop sub-sample. However, it is important to note that most points from the bulk and microlayer samples are in the baseline of the $\mu$l-NIPI experiment, and can therefore be seen as upper limits. It is notable however, that the 'microlayer' sample obtained with a glass plate had a lower activity than scooping the surface water, which might suggest that the ice active components may only have an intermediate affinity for the glass plate. Nevertheless, the fact that the upper layers of water in the AEGOR are enhanced in INP suggests that organic INP material scavenged by bubbles resides at the water surface and is likely surface-active (i.e. material which preferentially resides at an interface). As such, this material may be scavenged by the bubbling in the chamber and be preferentially aerosolised during the bubble bursting process.

### 3.2 Temperature regime below 248 K (Aerosolised samples)

The ice nucleation results of the AIDA and INKA measurements, expressed as ice nucleation active site densities versus temperature $n_s(T)$, are shown in Fig. 5 (SML samples) and Fig. 6 (algal cultures). With respect to the experiments where AEGOR was used for aerosol generation, some samples did not exhibit a detectable freezing signal above the background (SM100, SM10, and SML8) and are therefore not included. As a comparison to our data, Fig. 5 includes a recently published dataset consisting of field measurements of sea spray aerosols and laboratory data of particles released during an algae bloom generated in a marine aerosol reference tank (DeMott et al., 2016). Furthermore, we show a parameterisation of the temperature-dependent $n_s$ values for desert dust particles (Niemand et al., 2012).

The various SML samples show little variation at temperatures below 248 K when probed in the AIDA chamber, meaning that the SML samples all exhibited similar ice nucleation activity ($n_s$ of $10^9$ m$^{-2}$ at temperatures between 240 - 244 K) and the individual $n_s(T)$-curves of the AIDA measurements form a rather compact block of data (Fig. 5). One notable exception is the ASCOS high-molecular weight sample (ASCOS high mol. w., 5 kDa to 0.22 $\mu$m). Whereas the foam and < 5 kDa ASCOS samples fall into the range of $n_s$ values observed for the other SML and STN microlayer samples, $n_s$ for the high-molecular weight sample is about one order of magnitude higher. This agrees with the $\mu$l-NIPI observations, where this particular sample also proved to be one of the most ice active. The ASCOS high-molecular weight sample consists of the high molecular weight dissolved organic matter of the collected SML sample. More specifically, it was shown in Orellana et al. (2011) and Gao et al. (2012) that this sample mostly contained marine colloidal gels. This might lead to an enrichment of ice active organic material and explains the high ice nucleation activity of this sample. Note that this sample is highly concentrated. The size range of the filtration of the sample indicates that macromolecules are responsible for the freezing of the sample. Most bacteria, cell debris, etc. are likely to be removed by the ultrafiltration. The size distribution of the nebulised ASCOS high-molecular weight sample resulted in particles with the smallest diameters compared with other samples, which might have an influence on the ice nucleation activity as well since the chemical composition of sea spray aerosol is highly size dependent. This sample might consist of smaller particles with a larger organic mass fraction compared to the other samples. Other field samples that proved to be particularly ice active in the high temperature regime like SML5, however, do not show superior ice nucleation activity

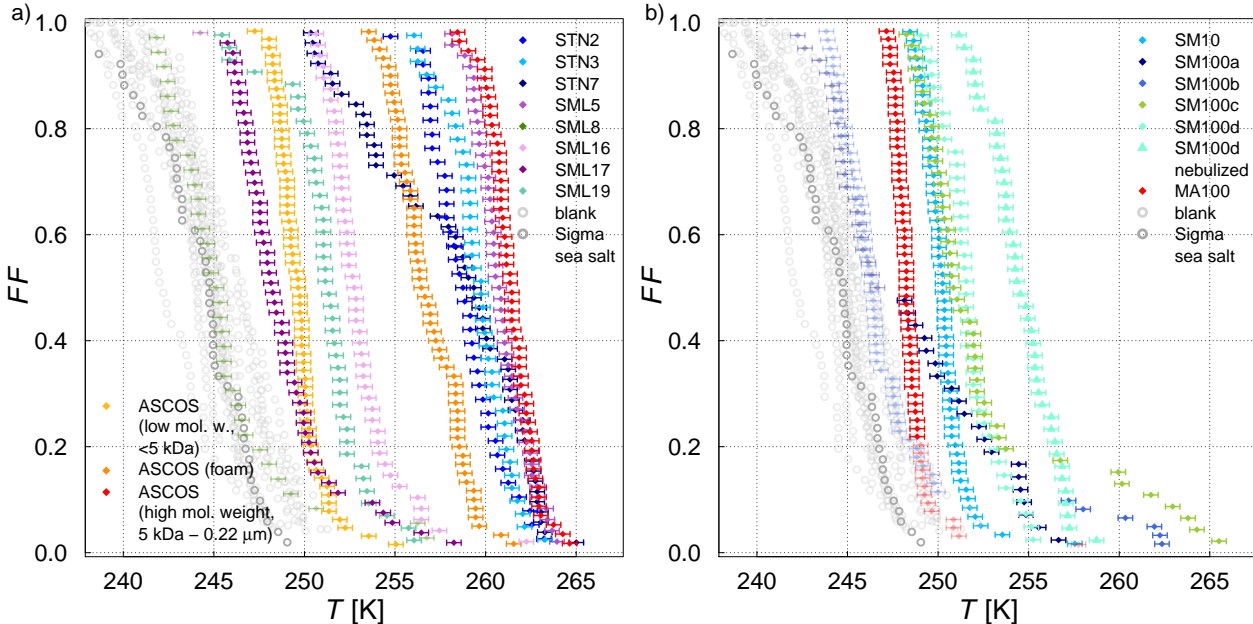

**Figure 3.** Fraction frozen curve, a measure of the fraction of droplets frozen at discrete temperatures, for:

a) 9 different SML field samples coming from three different Arctic field expeditions (ACCACIA, NETCARE, ASCOS) measured with the $\mu$l-NIPI (droplet freezing technique, bulk, undiluted samples). The field sample from ASCOS was treated in three different ways (see Sect. 2.1).

b) Two cultured diatom species measured with the $\mu$l-NIPI (droplet freezing technique, bulk): *Skeletonema marinoi* (SM) and *Melosira arctica* (MA). The SM sample was investigated for two different nutrient regimes (see Sect. 2.1). Two duplicate samples of SM100 (SM100a and SM100b) are reflecting the variability of the sample. SM100a and SM100b are from two bags collected from the same culture. SM100c is a sub-sample of SM100b after 2 months storage. SM100d, a sub-sample of SM100b, was in storage for 10 months, and was then nebulised and retested to determine the effect of the aerosolisation on the sample.

The points with reduced opacity represent upper limits for those data points, as they could have been affected by background signal.

Note that the temperature in both plots was not corrected for freezing point depression caused by salts because the water activity was not available for all samples.

at temperatures below 248 K. This is an indication that different types of ice active materials might cause the freezing in the different temperature ranges, an issue that will be further discussed in Sect. 3.3 when combining the AIDA and $\mu$l-NIPI data sets.

In order to facilitate the comparison of the AIDA measurements with previous studies of ambient marine aerosols, we chose to represent the DeMott et al. (2016) data in Fig. 5 by a grey shaded area that encompasses the observed range of nucleation site density values $n_s$. A similar representation was used by McCluskey et al. (2017), who have determined $n_s$ for nascent sea spray aerosol particles during phytoplankton blooms in the laboratory. These data are not separately depicted because they fall into the regime of the DeMott et al. (2016) dataset. A particular subset of the DeMott et al. (2016) data is highlighted in

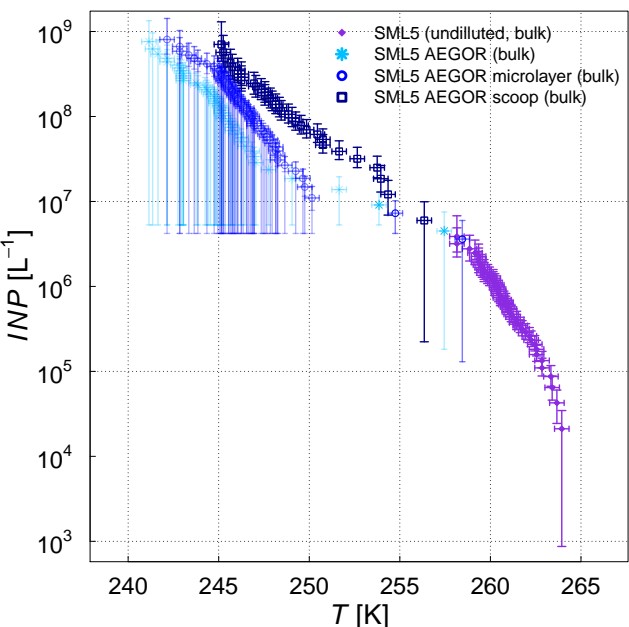

**Figure 4.** Cumulative INP concentration per unit volume field sample SML5 for the pure sample in comparison to different dilutions (sub-samples from AEGOR: bulk, microlayer, scoop; see text for details). Where the lower error bar is unchanged from the previous point, there may have been no additional INP detected above the background signal. Note that the temperature in this plot was not corrected for freezing point depression caused by salts because the water activity was not available for all samples.

Fig. 5 by the grey stars. These data points refer to a laboratory experiment in the Marine Aerosol Reference Tank (MART) following the peak of the phytoplankton bloom. The $n_s$ values derived from the AIDA measurements for the field samples fall into the range of former observations, albeit towards the upper, more ice active regime of the data by DeMott et al. (2016). The MART data for the artificially enhanced phytoplankton bloom is in good agreement with the upper thresholds of $n_s$ for our field samples. Given that most of the AIDA measurements were made by aerosolising the undiluted SML solutions with the

nebuliser, it can be expected that this dataset indeed represents an upper limit of the ice nucleation activity of natural sea spray aerosol particles.

The experiments where AEGOR was used for aerosol generation shed some light on how much of the ice active material in the SML bulk solutions may be released during the process of air entrainment, bubble scavenging and bubble bursting. For both sample types investigated, the algal cultures and natural SML samples, we find examples where the ice nucleation

activity observed of particles generated using the AEGOR tank remains similar to the ice activity of aerosols generated by nebulising the pure sample despite the strong dilution of the samples with artificial seawater in the AEGOR tank (SML5, Fig. 5; MA100, Fig. 6). This suggests that in some cases, the organic INP material is indeed preferentially scavenged by the bubbling in the seawater tank and aerosolised during the bubble bursting process. For other samples, however, the ice nucleation activity was reduced to below the detection limit ($n_s$ of $10^8$ m$^{-2}$) after the dilution in AEGOR (SML8, SM10, and SM100).

This variability in the AEGOR experiments might explain why the previous field measurements of sea spray aerosol particles show a huge spread in the $n_s$ values, whereas the laboratory nebuliser data fall into a narrow range at the upper end of the ice nucleation activity scale. Note that this upper limit of the ice nucleation activity of the field samples, however, is still one order of magnitude lower than the $n_s$ parameterisation for mineral dust [Fig. 5, Niemand et al. (2012)], underlining the relatively poor heterogeneous ice nucleation activity of sea spray aerosol particles compared to other atmospherically relevant types of INPs

in the temperature range above 248 K. In the (High) Arctic both transported dust and sea spray aerosol (transported or locally originated) can be present (see Willis et al. (2018) for a thorough review of literature). However, which source is dominant for ice nucleation might be locally very different. In regions dominated by sea spray aerosol the fraction of organic matter within the aerosol population is another uncertainty.

At low temperatures, the algal cultures had similar ice nucleation activities compared to the field samples, with *Melosira*

*arctica* being slightly more ice active than *Skeletonema marinoi*. For *Skeletonema marinoi* grown under replete and deplete nutrient conditions, the culture with the highest nutrient limitation and inhibited growth (SM10) had somewhat lower $n_s$ values compared to SM100 and SM60, but this trend is only distinct in the AIDA data and not as clearly visible in the INKA measurement. For comparison, we added previously published $n_s(T)$ values for two other algae, the diatom *Thalassiosira pseudonana* (Knopf et al., 2011) and the green algae *Nannochloris atomus* (Alpert et al., 2011a) [the data points were taken from Murray

et al. (2012)]. The ice nucleation activities of these two species are in reasonable agreement with the data presented here. They lie towards the lower end of the AIDA data and fully overlap with the range of the $n_s$ from the INKA measurements.

With respect to the comparison between the AIDA and INKA measurements, the INKA results tend to be shifted to lower $n_s$ values, although the INKA data partly overlaps with the AIDA data within the respective error bars. As previous INP measurements for insoluble aerosol particles such as soil dust have shown good agreement between AIDA and INKA (DeMott et al.,

2018a), the deviation for the current study with soluble, marine aerosol particles might be related to the particles' phase state. For soluble aerosols, the different time scales and particles' phase state evolution in the AIDA and INKA measurements might affect the observed INP data. In AIDA, the aerosol particles are initially suspended as aqueous solution droplets, gradually take up water when the expansion cooling run is started, are activated to $\mu$m-sized cloud droplets when the relative humidity exceeds 100%, and potentially nucleate ice by immersion freezing upon further reduction of the temperature during expansion

cooling. These processes occur on an overall time scale of approx. 5 min. For the INKA measurements, the aerosol particles are suspended as effloresced crystals in the APC chamber. During a very short time period of only 10 to 15 s in the first section of the CFDC chamber, the particles have to undergo the complex trajectory of deliquescence, droplet activation, and freezing. The short residence time in INKA might prevent equilibration of the aerosol to the instrument conditions. Thus, it is possible that at certain locations there is not enough water vapour present to fully activate the aerosol particles to cloud droplets and that

this effect may account for the slightly lower $n_s$ values compared to the AIDA measurements. Note that efflorescence might as well change the INP activity of the aerosol particles.

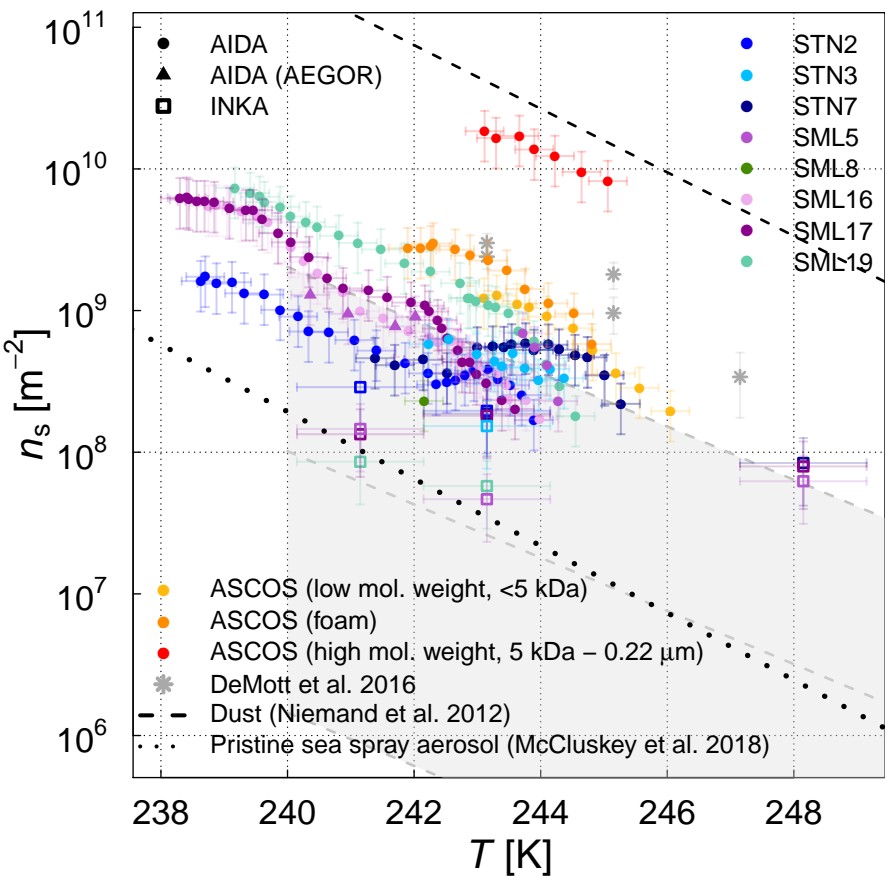

**Figure 5.** Surface active site density $n_s$ as a measure for ice nucleation activity at different temperatures for 11 different SML samples from the AIDA (coloured full circles and triangles) and INKA (open squares) measurements. The field sample from ASCOS was treated in three different ways (see Sect. 2.1). Different symbols show the different aerosolisation techniques for the AIDA measurement (nebuliser in circles, AEGOR in triangles). The AIDA $n_s$ data were corrected for the background ice nucleation mode observed in the reference experiments with purely inorganic Sigma-Aldrich sea salt solution droplets (see Sect. 2.4). The data of DeMott et al. (2016) is shown as a grey shaded area (fit and shifted fits to the upper and lower limit of the data) and grey stars (MART phytoplankton bloom), see text for details.

### 3.3 Combined bulk and aerosol phase measurements

One of the central aims of this study was to analyse the ice nucleation behaviour of Arctic SML samples and two different algal cultures over the full temperature range relevant for freezing in mixed phase clouds. We also wanted to assess if the ice nucleation material is transferred from the bulk to the aerosol phase. The samples were measured with different instruments sensitive to different temperature regimes: AIDA and INKA below 248 K (aerosol phase) and $\mu$l-NIPI above 248 K (bulk). Here we attempt to directly compare the AIDA and $\mu$l-NIPI datasets. The INKA dataset is not included in the comparison since

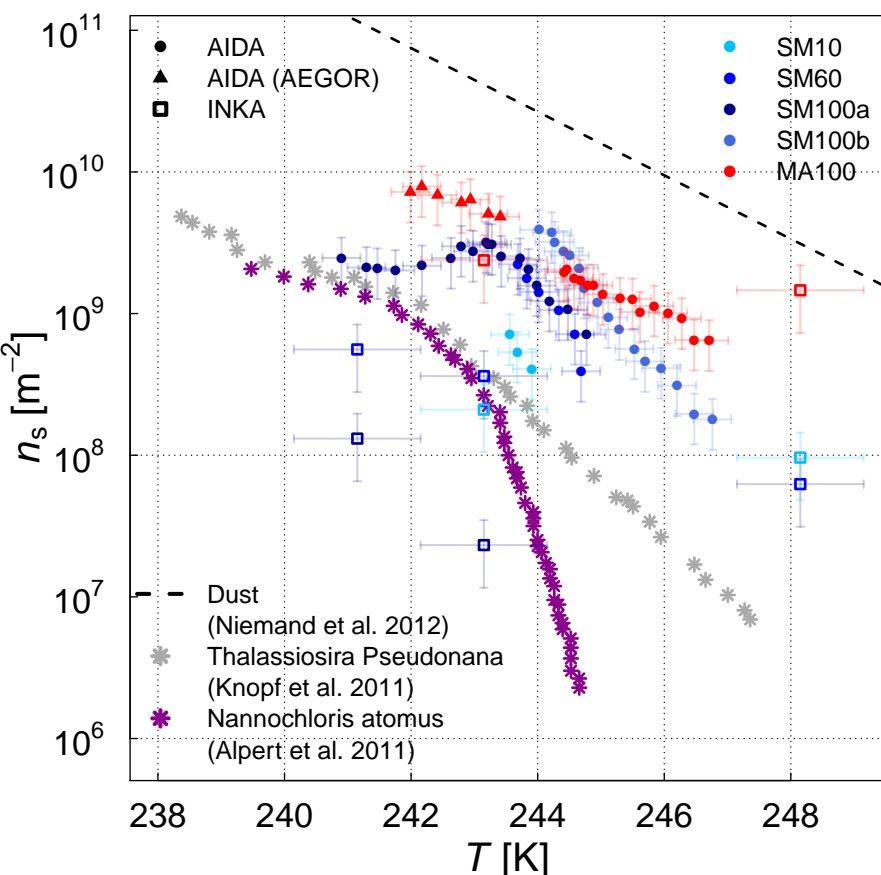

**Figure 6.** Surface active site density as a measure for ice nucleation activity at different temperatures for the two different diatom species (SM and MA) from the AIDA and INKA measurements. For SM, three samples grown under different nutrient regimes to generate cultures with different exudate properties (SM10, SM60, SM100) are shown. Literature $n_s$ data for *Thalassiosira pseudonana* and *Nannochloris atomus* are shown as a comparison. The AIDA $n_s$ data were corrected for the background ice nucleation mode observed in the reference experiments with purely inorganic Sigma-Aldrich sea salt solution droplets (see Sect. 2.4).

the AIDA dataset is more comprehensive and has a finer temperature resolution than the INKA data.

To enable comparison and answer the question if the ice nucleating material is transferred from the bulk to the aerosol phase, both datasets (AIDA and $\mu$l-NIPI) require normalisation so that the ice nucleation behaviour can be expressed with the same quantity as a function of temperature. We have chosen to normalise both sets of data to the mass of salt present in the solution droplets since this quantity can be estimated for both approaches. Thus, the ice nucleation behaviour is expressed as ice nucleation active site density per mass of salt ($n_m$; $[n_m] = g^{-1}$). It is more obvious how to treat and harmonise ice nucleation data using materials like mineral dust which have a relatively well-defined surface area. The surface area of an aerosol dispersion can be used to derive $n_s$ in much the same way as dust particles in bulk suspension. However, when the ice nucleating material



in a sample is soluble or forms colloidal suspensions then it is less clear how to treat it. This is especially complex for the marine system, where the bulk sample can be very different from what is aerosolised into the atmosphere - one question that we want to investigate a bit further by comparing the AIDA and the $\mu$l-NIPI datasets. While we can, and have, derived $n_\mathrm{s}$ values for the AIDA and INKA data where the surface area is the surface area of the dry aerosol, we cannot do this for the bulk suspension measurements from the $\mu$l-NIPI instrument. Similarly, while we have a measure of organic mass for the bulk microlayer samples we do not have a measurement of the organic mass in the aerosol phase, hence we cannot normalise to organic mass. Solution volume cannot be used, since the volume of the solution of the aerosol changes as its concentration alters to come to equilibrium with the chamber conditions. Hence, we have chosen to normalise to the mass of salt, a quantity which can be readily estimated from both the bulk and aerosol experiments. When contrasting the resulting $n_\mathrm{m}$ values it should be borne in mind that the spread in activities is likely an indication of the range of concentrations of the ice active components as well as variability in the activity of those components. The objective of our work was to compare droplet freezing assay results with aerosolised measurements, rather than to derive a quantity which could be used to predict atmospheric INP. Ideally, we would quote active sites per unit mass of the nucleating component, but if the identity and mass of the nucleating component is unknown this is not possible (as in this case). However, this approach enables us to investigate if the bulk and the aerosolised samples behave similarly and if both ice nucleation techniques complement each other when normalised and brought into one context.

For the $\mu$l-NIPI data we derive the salt concentration for each sample in g/L using the measured water activity of the samples and the parameterisation linking the water activity and salt concentration of seawater presented by Tang et al. (1997). To calculate the ice nucleation active site density per mass of salt, the measured INP/L is simply divided by the salt concentration in g/L. For the samples where no water activity was measured as part of this study (see Table 3), the values from Wilson et al. (2015) (for the ACCAIA SML samples) or an average of all SML samples (for the NETCARE STN samples) was used. We added an additional uncertainty of 20% (arbitrary) to the error bars for the $n_\mathrm{m}$ values of the samples where the water activity was not directly measured. The ASCOS samples are not included in the unified dataset. Their water activity could not be directly measured because the remaining sample volume was too small. Furthermore, these samples were treated differently to the other microlayer samples so an average water activity might not be a good representation for theses samples.

For the AIDA data the measured $FF$ was normalised with the measured mass concentration of dry particles (as obtained from the SMPS and APS measurements, see discussion in Sect. 2.2), instead of using the particles' surface area concentration for normalisation that yielded the $n_\mathrm{s}$ data shown in Figs. 5 and 6. The underlying assumption is that the dominating constituents in terms of mass is salt with a density of $2.017 \pm 0.006$ g cm$^{-3}$ [Sigma-Aldrich sea salt; Zieger et al. (2017)]. Considering the composition of marine aerosols as presented in Gantt and Meskhidze (2013) this assumption is fair for the typical sizes of aerosol particles aerosolised into AIDA.

**INP transfer from the bulk to the aerosol phase**

The combined ice nucleation activity of the field samples is shown in Fig. 7. The combined temperature spectra for the ice nucleation activity of the algal samples is shown in Fig. 8 and Fig. 9; the samples were split in two figures for clarity.

We first turn to the comparison between the AIDA and $\mu$l-NIPI measurements for the algal and field samples focusing on the difference between aerosolised and bulk samples. A significant difference between the AIDA and $\mu$l-NIPI measurement is that one is derived from an aerosolised sample and one is derived directly from the pipetted culture medium. Comparison between $\mu$l-NIPI, AIDA and other instruments in a recent intercomparison was very good (DeMott et al., 2018b). Inspection of the data in Fig. 7, Fig. 8 and Fig. 9 suggests that the data from the two techniques might be consistent, but $n_{\mathrm{m}}$ would have to be extremely steep at the intermediate temperatures. The discontinuity of the AIDA and the NIPI data, i.e. the shift of the AIDA data to higher $n_{\mathrm{m}}$ values might be related to a change of physical characteristics upon aerosolisation. Aerosolisation may alter the physical characteristics of the ice nucleating material compared to when it is in the culture medium through breaking up aggregates or disrupting cells. This was shown for *Pseudomonas syringae* cells in the study of Alsved et al. (2018). Hence, it is feasible that the ice nucleation activities of the aerosolised samples in the AIDA experiments are higher than those in the $\mu$l-NIPI experiments. However, there is a recognisable difference between both types of samples. The aerosolisation technique might exert more of an influence on the cultured samples compared to the microlayer samples, where the INP are thought to be associated with submicron organic detritus, rather than intact cells. For the SML samples, it is therefore reasonable to assume that the composition of the aerosolised solution droplets probed in the AIDA chamber is very similar to that of the corresponding bulk solutions used in the $\mu$l-NIPI measurements. Indeed, the $n_{\mathrm{m}}$ spectrum looks more uniform as compared to the algal cultures. Most samples feature a rather continuous slope in the temperature-dependent INP spectrum. One notable exception is the STN7 sample, which shows a pronounced, step wise change in the ice nucleation behaviour at about 263 K.

For the algal cultures the assumption that the aerosolised and bulk samples are similar is not necessarily valid. In order to investigate if the process of nebulising influences the ice nucleating activity of cell suspensions, we nebulised a SM100 sample, collected the nebulised sample as a bulk liquid and retested its ice nucleating activity using the $\mu$l-NIPI. Nebulisation increased the activity of the sample (see Fig. 3). We suggest that this might be consistent with the break up or rupture of cells in the vigorous nebulisation process, which might then release macromolecular ice nucleating materials. Alternatively, there might be agglomerated cells or colloidal particles inside the sample. That means that ice active sites can be either inaccessible or simply concentrated in a few particles. These aggregates might remain relatively intact during pipetting, but may be disrupted on nebulisation. It would have the effect of dispersing the ice nucleating entities throughout the aqueous suspension, thus increasing the probability of freezing across the droplet distribution when nebulising the sample. However, nebulising MQ water (not shown) showed that some impurities can likely be introduced by the nebuliser itself. These hypotheses deserve further investigation in the future.

Further to this, we have the hypothesis that the aerosolised material entering AIDA was very different compared to the pure cultures. For example, first analysis of electron microscopic pictures of aerosol particles contained in AIDA (representative for particles aerosolised with a nebuliser into AIDA) during the experiments with *Skeletonema marinoi* showed no cells or

obvious cell fragments visible (see left picture of Fig. 10). This is consistent with the microlayer being dominantly composed of organic detritus and might be a result of biochemical processes within the microlayer. In contrast the right picture of Fig. 10, where SM100 droplets were pipetted directly from the solution, shows clearly cells, which are then also present in the droplets analysed with $\mu$l-NIPI. However, a more detailed analysis would be needed to give a final answer on the difference of the aerosol particles in AIDA compared with aerosol particles within pipetted droplets.

**Dilution tests bulk measurements**

Figure 8 shows the $n_m(T)$ spectra for the SM100 culture and the variability including two SM100 samples (a and b for biological variability; c and d for storage effects) as discussed in section 3.1. The latter (Fig. 9) shows the spectra for MA100 and SM10. To bridge the gap in the ice nucleation spectra between the AIDA and the $\mu$l-NIPI data, we did additional dilution experiments with $\mu$l-NIPI to extend the temperature regime of the $\mu$l-NIPI data to lower temperature. Diluting the SM100 and MA100 sample has the effect of reducing the freezing temperature and increasing $n_m$. Thus the curves from the undiluted samples can be extended to lower temperatures. That works well for SM100 and partly also MA100. For MA100 the slope of the $n_m$ curve continues to be steep throughout the dilutions. However, there are some points which may have been affected by the background signal, which are denoted by the larger lower error bar value. It is not clear why there is such a difference in the behaviour after dilution between the SM100 and MA100 samples, and further investigation into the differences in their composition and how this is related to their ice nucleating ability is necessary.

**Temperature dependent difference in ice nucleation behaviour**

As a striking result, there is much more variability in the ice nucleation activity of the samples when analysed with the $\mu$l-NIPI than with AIDA (approx. 15 K vs. 5 K). This larger variability in the high temperature range has been observed in other studies, too, e.g. for soil or agricultural dust (O'Sullivan et al., 2014; Schiebel, 2017; Suski et al., 2018). One explanation for this behaviour could be that there are multiple INP types in seawater, just like there are in terrestrial samples, leading to a high diversity of the INP spectra at high temperatures. At low temperature the ice nucleation activity is much less variable and low throughout all samples.

## 4  Conclusions

In this study the ice nucleation activity of several bulk and aerosolised SML samples from the Arctic region was investigated and compared with pure and aerosolised samples of two diatom cultures (*Skeletonema marinoi* and *Melosira arctica*). The measurements were conducted with a suite of ice nucleation instruments (AIDA, INKA, $\mu$l-NIPI) which are sensitive in different temperature regimes across the whole mixed-phase cloud temperature range (below and above 248 K). In order to compare the different approaches and the ability of the ice nucleating material to transfer to the aerosol phase we have normalised all of the measurements by the salt mass present in the bulk and aerosolised samples. Normalisation in this manner results in an ice nucleation active site density per mass of salt $n_m$.

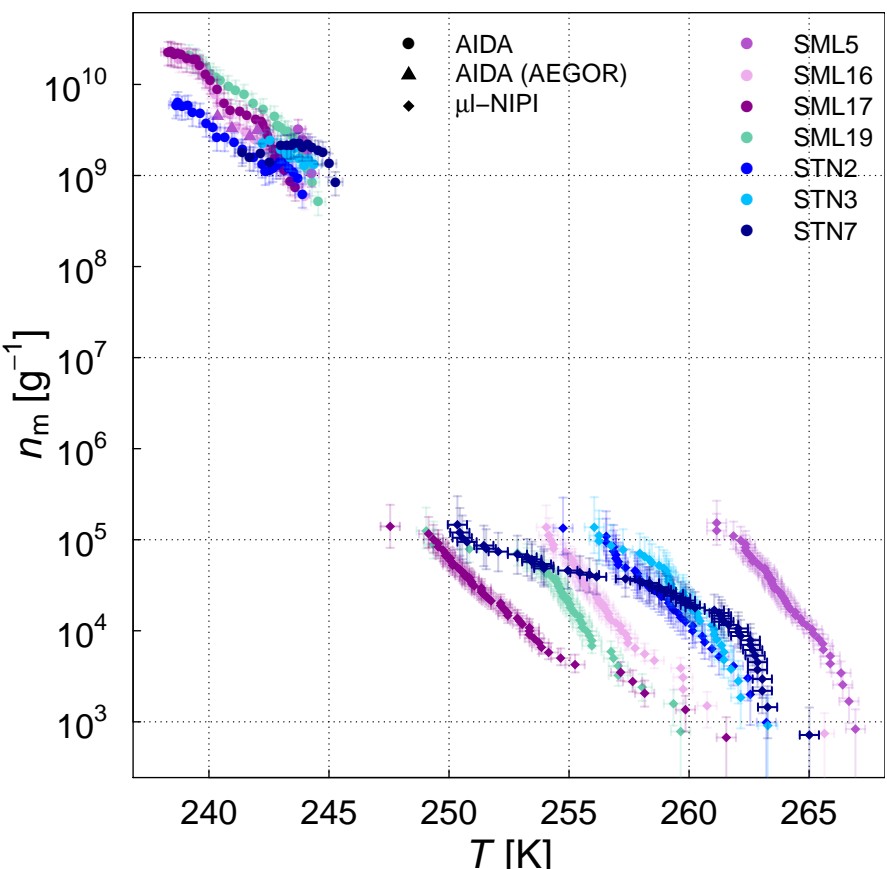

**Figure 7.** Normalised AIDA and $\mu$l-NIPI measurements for 7 field samples showing a full ice nucleation spectrum represented as ice nucleation active site density per mass of sea salt $n_\mathrm{m}$. For the AIDA measurements both aerosolisation techniques (nebuliser and AEGOR) are included. The points of the $\mu$l-NIPI measurements which could have been affected by background signal and represent upper limits are indicated by a lower error bar that is unchanged from the previous point, as there may have been no additional INP detected above the background signal. The temperature in this plot was corrected for freezing point depression caused by salts for the $\mu$l-NIPI measurements.

Our three main objectives were: first, the comparison of the ice nucleating ability of two common phytoplankton species with Arctic microlayer samples, second, the impact of the aerosolisation technique on the results, and third, the sample variability over the entire mixed-phase cloud temperature range. Concerning these objectives we can draw the following conclusions:

When comparing the full temperature spectrum of the algal cultures with the field samples it is evident that the culture samples are similar to the field samples in the low temperature regime but are not among the most ice active samples of the spectrum in the high temperature regime. As the investigated algae species show less ice activity in the temperature regime above 248 K compared to the natural field samples, we conclude that they, especially *Melosira arctica*, cannot explain the freezing at the high temperatures. A normalisation of the samples to the atmospheric algal content would be needed to quantify

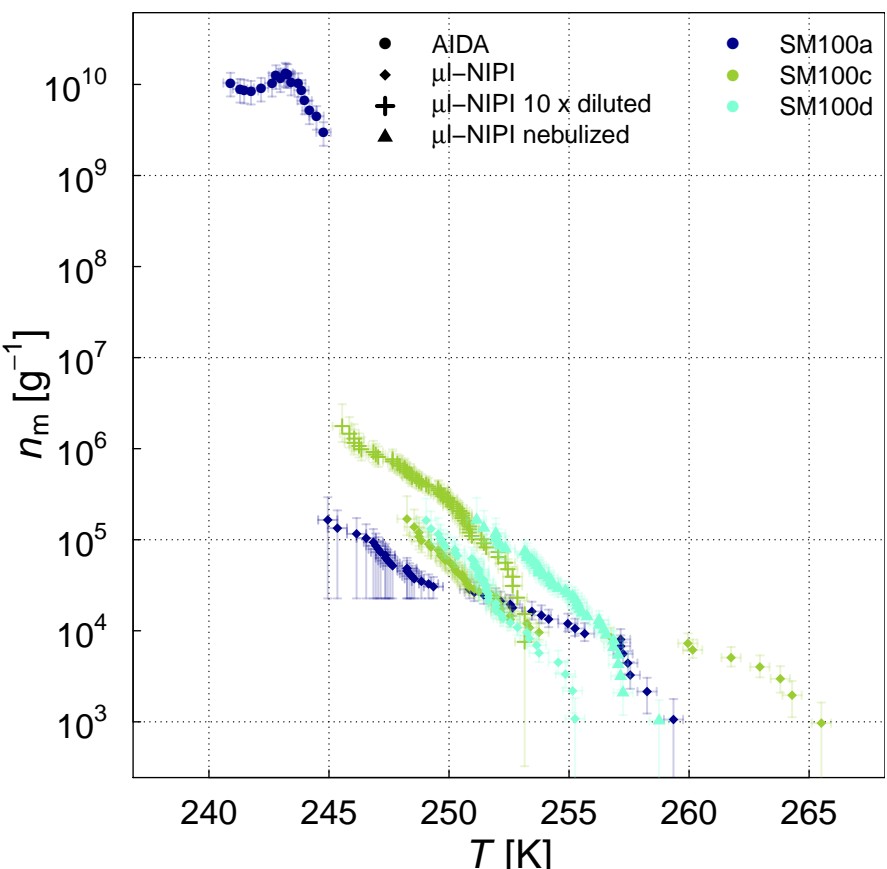

**Figure 8.** Ice nucleation active site density per mass of sea salt $n_{\mathrm{m}}$ estimated from the AIDA and $\mu$l-NIPI measurements for the SM100 culture samples. For the $\mu$l-NIPI measurements, the SM100 samples were additionally diluted with ultrapure water. Note that the dilution was conducted up to 8 weeks after the main campaign (Leeds, UK). The points of the $\mu$l-NIPI measurements which could have been affected by the background signal and represent upper limits are indicated by the lower error bar unchanged from the previous point, as there may have been no additional INP detected above the background signal. The temperature in this plot was corrected for freezing point depression caused by salts for the $\mu$l-NIPI measurements.

this observation. This result indicates that the INPs active at the highest temperatures are not one of the two types of phytoplankton cells studied or their exudates. However, since we have only tested two mono-species grown axenically and harvested at exponential growth phase, we cannot rule out ice nucleation being triggered by a consortia of microorganisms facilitating break-up of cells and mass-release of organic matter from a phytoplankton bloom. The freshly produced pure algae cultures are different from the diluted field samples, which are highly diverse in terms of composition. Aged algal cultures may exhibit

a different freezing behaviour. For *Skeletonema marinoi*, the culture was grown at different nutrition conditions to test the dependence of the freezing on the algal characteristics, such as total organic carbon (cell organic carbon and all dissolved organic

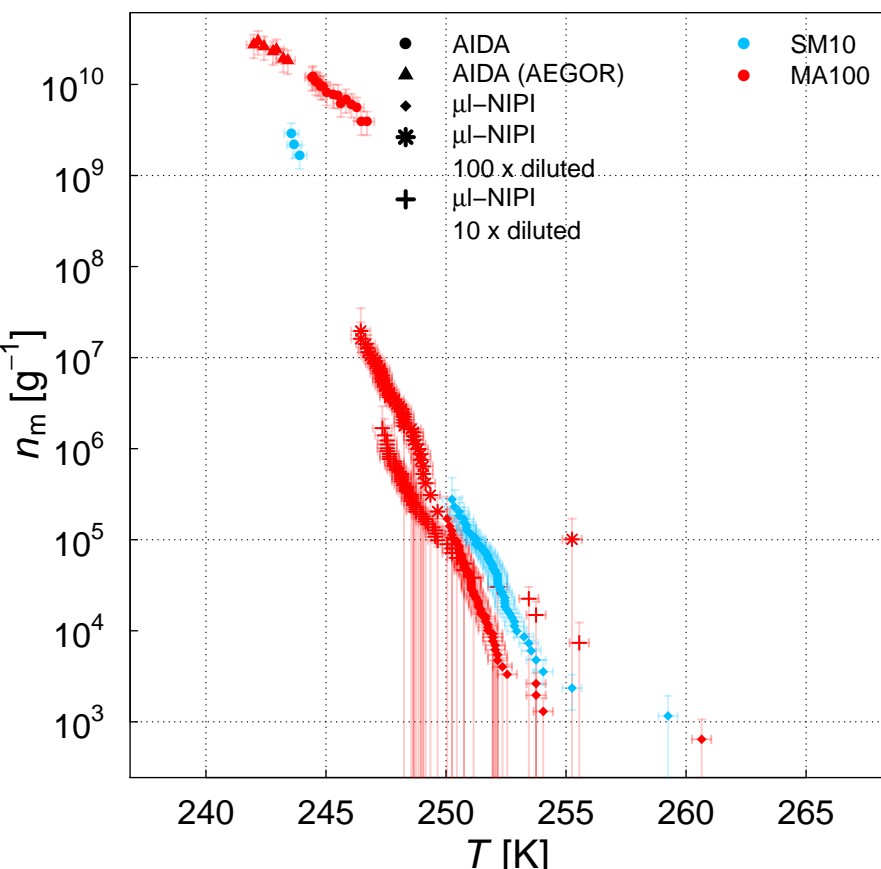

**Figure 9.** Ice nucleation active site density per mass of sea salt $n_{\mathrm{m}}$ estimated from the AIDA and $\mu$l-NIPI measurements for the SM10 and MA culture samples. For the AIDA measurements both aerosolisation techniques (nebuliser and AEGOR) are included. For the $\mu$l-NIPI measurements, the MA100 sample was additionally diluted with ultrapure water. Note that the dilution was conducted up to 8 weeks after the main campaign (Leeds, UK). The points of the $\mu$l-NIPI measurements which could have been affected by background signal and represent upper limits are indicated by the lower error bar unchanged from the previous point, as there may have been no additional INP detected above the background signal. The temperature in this plot was corrected for freezing point depression caused by salts for the $\mu$l-NIPI measurements.

carbon), cell wall structure, colony length etc.. No significant difference could be found when comparing the ice nucleation behaviour of the samples grown at different rates and under varying nutrient limitation, so there is no clear evidence for a correlation between the total organic carbon content of the culture sample (see Table 3) and the freezing of the sample.

A key aspect of this study is that we have used both a sea spray simulation chamber and a nebuliser to introduce samples into AIDA (low temperature regime). Using a sea spray simulation chamber (AEGOR) allowed us to test the effect of mimicking the process of bubble bursting on the ice activity of the aerosol generated. A larger spread was observed in general for the SML samples diluted in AEGOR - some retained the activity of the undiluted sample, in some cases the IN ability decreased below

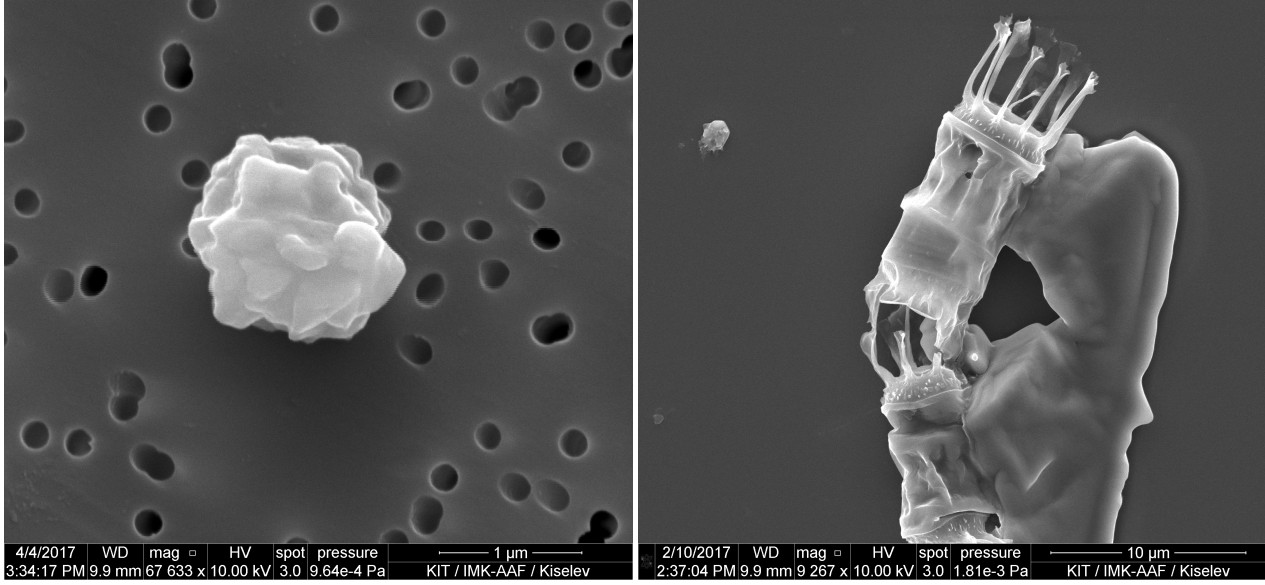

**Figure 10.** Electron microscope pictures of SM10 (aerosolised by AEGOR) collected from AIDA (left) and SM100 in droplets pipetted directly from the solution (right).

the detection limit. Lower ice nucleation active site densities (for the cases where the IN ability decreased below the detection

limit) can be explained by the difference in the size distribution of the aerosols generated by the two approaches.

Analysing the ice nucleation spectra over the whole temperature regime, the SML field samples exhibit a high variability in ice nucleation activity in the temperature regime above 248 K compared with lower temperatures. Above 248 K the variation in the median freezing temperature $T_{50}$ is approx. 15 K with some samples showing a strong freezing signal at high temperatures ($T_{50} \approx 262$ K), while below 248 K the spread of $T_{50}$ is only approx. 5 K. The behaviour of the samples in the different

temperature regimes might be related to different types of INP active in the different regimes. In the temperature range below 248 K the results of this study are in the upper range of the values measured by DeMott et al. (2016), which show a larger spread compared to the results of the nebulised samples. This larger spread could be explained by the aerosolisation (see paragraph above). However, neither the SML nor the algal samples exhibit a strong freezing signal in the low temperature regime (below 248 K) compared to desert dust. There was no significant freezing above the detection limit in the AIDA chamber (around

$2 \times 10^8$ m$^{-2}$) at temperatures higher than 246 K. The ice nucleation active surface site densities were generally at least one order of magnitude lower than those for desert dust.

We also tentatively show that nebulisation enhances the ice nucleating ability of some cell cultures. We suggest that the aerosolisation process using a nebuliser might rupture individual cells allowing ice nucleating macro-molecules to be dispersed through the aerosol population. Alternatively aggregates of cells or colloidal material may be broken up during aerosolisation.

This might be unlikely to be relevant for environmental conditions. Here, this may lead to the aerosolised samples in the AIDA chamber having a greater ice nucleating activity than they would otherwise have. Pipetting of droplets, as done for the $\mu$l-NIPI

measurements, might be much less likely to exert sufficient force on the samples to break up cells, aggregates or colloidal material. Our hypothesis is that this process is particularly important for cell cultures and is less important in microlayer samples which consist of organic 'detritus' rather than intact cells (i.e. the organic material is already well dispersed).

In the experiments with microlitre volume droplets ($\mu$l-NIPI), which are sensitive to rarer ice nucleating particles, some of the SML samples have values of $T_{50} \approx 262$ K. This indicates that there is a low concentration of relatively active ice nucleating entities in these samples. The high variability observed in the high temperature regime suggests that there is a substantial variability in the presence of INP in the samples. What gives rise to this variability and what factors control it is a particularly important outstanding question. Previous work has shown that both the type and concentration of INP varies substantially

throughout the development and decay of a phytoplankton bloom (Wang et al., 2015; McCluskey et al., 2017; Wilbourn et al., 2020). There are perhaps various types of marine INP from different biological sources present in these natural samples. While our results, and those in the literature e.g. Knopf et al. (2011); Alpert et al. (2011a), show that phytoplankton can nucleate ice, it is also feasible that bacteria exploiting organic detritus from a plume might nucleate ice (Fall and Schnell, 1985). The presence of bacterial proteinacious ice nucleating material would be consistent with the observation that INP in microlayer samples

are heat sensitive, e.g. Wilson et al. (2015); Irish et al. (2017). However, a heat treatment test (see appendix) on SM100 did not give a clear indication for this hypothesis: in the low temperature regime no heat sensitivity of freezing, but a significant deactivation of freezing on heating could be seen in the high temperature regime for that sample. Since bacteria tend to be larger than 200 nm and bacterial ice active proteins are cell-membrane bound, one would expect to loose the ice activity associated with bacteria when filtering the sample through a 0.2 $\mu$m filter (Maki et al., 1974; Murray et al., 2012). This could not be

seen in our results of the differently treated ASCOS samples, where the filtered ASCOS sample < 0.22 $\mu$m did not show any reduction in ice nucleation activity. The ice nucleation activity of this sample indicated that macromolecules are responsible for the freezing, which were highly concentrated in the sample. It was suggested in literature that marine INP are very small and heat sensitive (Schnell and Vali, 1975; Vali et al., 1976; Wilson et al., 2015; Irish et al., 2017, 2019b), which is consistent with an ice nucleating protein responsible for the INP activity, similar to those found in terrestrial fungi (Pouleur et al., 1992;

O'Sullivan et al., 2015, 2016). Marine viruses may also fit this size requirement, although we are not aware of any studies on them for ice nucleation. A different candidate could be bacterial vesicles which are 50 - 200 nm particles and can retain the ice nucleating activity of their parent bacterium (Phelps et al., 1986). Another possibility is that the ice nucleating ability of the organic material in seawater is in part due to riverine input. River water is known to harbour large quantities of macromolecular INP (Larsen et al., 2017; Moffett et al., 2018) and the observed anti-correlation between INP and salinity is consistent with

a significant riverine input of INP to some marine environments (Irish et al., 2019b). Given the massive diversity of the high temperature INP observed in seawater in this and previous studies, e.g. Schnell and Vali (1975); Schnell (1977); Wilson et al. (2015); Irish et al. (2017, 2019b), it is likely that the sources of these INP are also highly variable and heterogeneous, much as they are in the terrestrial environment.

     From our study it is difficult to answer the question whether Arctic regions may have local marine sources of INPs and how

much they influence Arctic mixed-phase clouds. At temperature above 248 K the ice nucleation activity of the investigated samples was very diverse, with some samples reaching a quite high median freezing temperature of 262 K, thus potentially

being able to trigger freezing in Arctic mixed-phase clouds. The measurements in the temperature regime below 248 K on the other hand did not show that the samples were particularly ice active, especially when compared to dust, despite the fact that the results show an upper limit for $n_s$. Both measurements differentiated in the way the samples were analysed (bulk vs. aerosol phase). This was most relevant for the cultured samples, giving some hint that aerosolisation of cell cultures may change the ice nucleation activity of these, a process that could be important in the environment as well.

## Appendix A: Heat test SM100

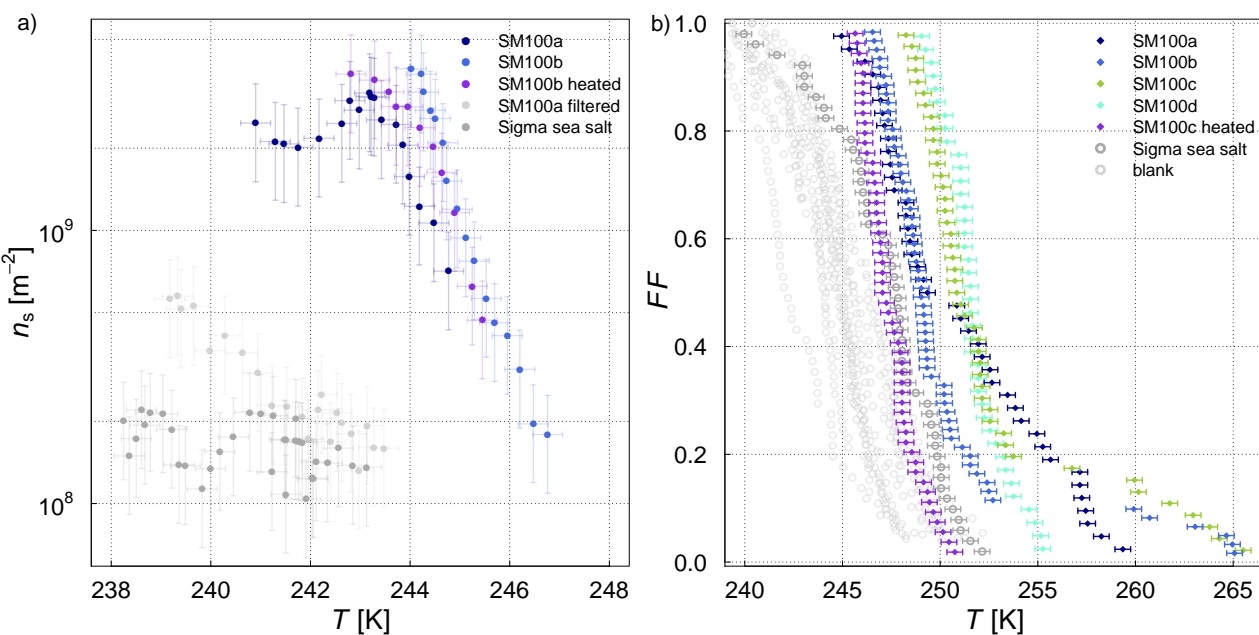

**Figure A1.** Heat test of SM100, measured with a) AIDA (Surface active site density $n_s$ as a measure for ice nucleation activity at different temperatures) and b) $\mu$l-NIPI (Fraction frozen curve). The temperature in the $\mu$l-NIPI plot was corrected for freezing point depression caused by salts.

*Code and data availability.* The data will be available at the KITopen data repository (https://www.bibliothek.kit.edu/cms/kitopen.php).

*Author contributions.* **Conceptualisation measurement campaign**: MS, RW; **Measurement campaign**: MS, RW (overall lead); RW, LI
(AIDA); GCEP, MPA (NIPI); SB, KH (INKA); SC (CCN); AAK, LI (ESEM); **Samples**: EG (algal cultures), GCEP (SML), CL (ASCOS); **Analysis of the data**: LI, RU, RW (AIDA); GCEP, MPA (NIPI); SB, KH, TS (INKA); SC (CCN); AAK (ESEM); **Visualisation**: LI; Fig. 1: LI, MS; Fig. 10: AAK; **Writing**: LI, MS, RW, KH, BJM, GCEP; **Review & editing**: All authors.

*Competing interests.* The authors declare that no competing interests are present.

*Acknowledgements.* We gratefully acknowledge the support of the Engineering and Infrastructure group of IMK-AAF, in particular Olga
Dombrowski, Rainer Buschbacher, Tomasz Chudy, Steffen Vogt, and Georg Scheurig. This study was supported by the Helmholtz-Gemeinschaft
Deutscher Forschungszentren as part of the program "Atmosphere and Climate". We thank Victoria E. Irish for shipment and help with the
SML samples (STN). We thank Nadine Hoffmann (IMK-AAF) for preparing the SM samples for the ESEM measurements. We thank
EUROCHAMP-2020 for TNA (Trans-national Access) support and funding and the Bolin Centre for Climate Research for supporting our
data workshop held in Stockholm in 2017. LI was supported by the Swiss National Science Foundation (Early Postdoc.Mobility) and the
Swedish Science Foundation (Vetenskapsrådet), with grant number 2015-05318. MES was supported by the Swedish Science Foundation
(Vetanskapsrådet) with grant number 2016-05100. CL was supported by the Swedish Science Foundation (Vetanskapsrådet) with grant number 2016-03518. BJM acknowledge the European Research Council (MarineIce, 648661), MB and SC Aarhus University and Hakon Lund
Foundation and AKB the Natural Sciences and Engineering Research Council of Canada for funding. We thank the two anonymous reviewers
for valuable feedback.

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
