# Peer review of "The ice nucleating activity of Arctic sea surface microlayer samples and marine algal cultures"

_Atmospheric Chemistry and Physics, 2020_

## Referee Comment (RC1) · Anonymous Referee #1 · 19 May 2020

Review of "Arctic marine ice nucleating aerosol: a laboratory study of microlayer samples and algal cultures" by Ickes et al., submitted to ACPD

The manuscript is a very well written and quite exhaustive study about the ice nucleation ability of different samples related to Arctic sea water (algal cultures and sea surface microlayer samples), examined with different measurement approaches. Measurement approaches include the examination of liquid samples on a cold-stage, and measuring aerosolized samples with the expansion chamber AIDA and the in-situ instrument INKA (a continuous flow diffusion chamber). For aerosol generation, two different methods were deployed, aerosolization and a plunging jet sea spray chamber.

[Figure]

Measurements were all done with great care and with up to date knowledge. The work describes and summarizes all measurements well, observed results are interpreted broadly and with great care. I particularly liked section 3.3, in which all different methods are compared, based on a normalization wrt. the salt content of the samples (and again with sufficient care, in that caveats of using this normalization are mentioned as well).

Having said all these positive things, I did miss some final information about what was learned for the atmosphere from this study. It is clear that the contribution of "real world sea spray" to atmospheric aerosol and particularly to the INP fraction is a different topic. But the authors decided to submit their work to ACPD, and in that context it will benefit from some summarizing remarks about the applicability of results from this study for atmospheric implications.

Additionally, although I have no big concerns about this work, there is a rather longish number of remarks I give below. They are all not essential, but should be dealt with to make this already good work such that it then can be published in ACP.

Specific remarks:

line 37: "The types of aerosol particles that constitute good INP are uncertain (DeMott et al., 2010)." As the field of atmospheric ice nucleation research had a VERY strong revival in the last decade, this sentence is a) not really correct, and b) the citation is quite old. Since then, there were several review papers, and there is already an older one (Szyrmer & Zawadzki, 1997; Hoose & Möhler, 2012; Murray et al., 2012; Kanji et al., 2017). I don't want you to cite all of them, it's just to show you why I think this sentence needs revision (or deletion).

line 42: "sea spray aerosol could be an important source of INP" – again, it can't be expected that you give a complete review here, but as this is a focus of your work, I wanted to point out these papers on the topic: McCluskey et al., (2018a, b) and Creamean et al. (2019).

line 48: "three main groups" - There is actually a quite new paper in which SML and airborne concentrations were connected (including cloud water): Gong et al. (2020). For air masses that were continentally and/or mineral dust influenced (Cape Verde), INP concentrations in the SML did not explain atmospheric INP concentrations - that does not say anything about the remote oceans, but is a piece in the puzzle, nevertheless, which is worth mentioning, as this helps closing a gap between ocean and atmosphere.

lines 84/85 and line 95-97: You write: "Another goal of this study was to improve our understanding of whether Arctic marine regions may have local sources of marine INPs." and "Through comparison of the ice nucleation activity of artificial seawater containing Melosira arctica with that of the SML samples we aim to shed light on how representative relevant algal cultures are for Arctic marine INP." - Comments on that in the summary (that I proposed above) would be highly welcome, although it is clear that it is not straightforward to draw conclusions on atmospheric concentrations of sea spray aerosol from artificial lab work. But there are results you can summarize!

Table 1: As you try to give an overview here, including the papers I referred to above makes sense (McCluskey et al., 2018a,b; Creamean et al., 2019; Gong et al., 2020). And there is one more for coastal Mexico by Ladino et al., (2019) which may fit.

line 323: "sample is well mixed, so that particles are distributed uniformly, and each droplet is representative." Actually, if the INP concentrations are so high that this is true, then each well freezes at the same time, yielding a super-steep freezing curve. When the sample is then diluted or is already more diluted to begin with (so that conditions are those for which measurements typically are made), a less steep increase of FF with decreasing temperature is observed. But then, strictly speaking, INP are Poisson-distributed and this here does not apply any more. Therefore, you could say "... so that particles are distributed randomly."

lines 385-389: Due to the dilution that was done, comparing FF does not make sense,

and the figure could have only been shown for INP concentrations (normalized) – at least a FF figure cannot be interpreted in this way. Please revise.

lines 408/409: This sentence here gives a wrong impression. It becomes clear in the next chapter (3.3), that there really is a lower variability for the AIDA data at the lower temperatures. But before normalizing data from the different instruments (as you do, based on the sea salt concentration below), you should refrain from any kind of comparison and discussion thereof.

lines 441-442: This conclusion confuses me a little. You used the dry(!) particle number size distribution to derive the surface area for both particle generation methods when you normalized. This basically means that the contribution of different aerosol particle types (such as particles consisting purely of salt, or of having a mix between salt and organics, ...) to the overall aerosol was similar, independent of the particle generation. And therefore, this conclusion does not hold.

line 449: And of course, for atmospheric relevancy, also the abundance of these different particle types has to be accounted for. Mentioning this here would be good.

lines 535-536: "the diluted sample having higher nm values compared to the undiluted sample in the same temperature regime." This may be an indication that the background was hit, i.e., one already measures background, dilutes more and again only measured background, but normalizes to a higher dilution. Check if this is the case here, and if yes, omit the data.

lines 582-583: To be able to draw this conclusion, results would have to be normalized to "atmospheric algal content", which wasn't done and (as you argue above) is difficult, even amongst the NIPI data. This should be mentioned.

Technical comments:

The word "freezing depression" is used consistently. "freezing point depression" is the more correct term, right? Same for "aerosolisation" -> "aerosolization"?

Figure 1: When initially looking at this figure, I wondered about the meaning of the arrow on the left (in the middle), going from the plunging jet tube to the droplet freezing experiment. It became clear later on. It's probably a matter of taste to leave it here or to delete it – I just wanted to point out my initial confusion.

lines 137-139: Doesn't the nutrient content determine the growth rate? - It puzzled me that you can somehow set things such that you get high growth with low nutrient content (middle case). Or is the growth determined separately, and you just already give these observations here? Then that should be made clear.

line 215ff: Upon reading this for the first time, I was confused about the influence this low temperature in AIDA during the preparation phase would have on the measurements. This is nicely explained later, and it would be good to point out that an explanation on the reason for choosing such a low temperature will follow.

line 247: "The smaller ... vessel" could better be introduced here as "A smaller 3.7 m3-sized stainless steel vessel located in the vicinity of AIDA ...".

line 306: "sample air flow is sheathed by dry particle free synthetic air" – that is only true initially - water vapor diffuses through the sheath air flows and the sample flow from one plate (the warmer) to the other. Consider reformulation.

line 371: Concerning a difference in ice nucleation temperature observed for different cooling rates, the papers you mention show that the influence of the cooling rate is rather small. Please add that explicitly! As you mention, this can rather not explain the differences you observe.

line 377: Concerning the loss of ice activity during storage, there is a paper on that for Snomax (Polen et al., 2016).

Figure 3: 1) The different shades of bluish green in b) are difficult to distinguish, engraved by the opacity changes. Maybe additionally also change the symbol styles between samples? Or use a broader range of colors? (Although it is nice how you use

consistent colors throughout the manuscript for the separate samples.)

Figure 3: 2) Also, check the legend in panel b): SM 100d appears twice, SM 100b not at all.

Figure 3: 3) "Two duplicate samples of SM100 (SM100a and SM100c)" - do you refer to the two bags in which the sample was delivered? In the text these are "a" and "b", while "c" is the one that was stored. Check and homogenize.

line 417: Delete "of" (at "contained of ... gels") or replace "contained" by "consisted".

Fig. 5, 7, 8, and 9: Maybe use a separate legend explaining the different symbol types (AIDA, INKA, ...), as it was done for Fig. 6. - It's a bit confusing to see one entry in the legend saying "SMLx", and then another sample is "AIDA".

line 539: Add "is" between "this" and "related".

line 561: Shouldn't "since" rather be replaced by an "although"? If it's assumed that INP are small, and not intact cells, aerosolization might not do additional harm. In fact, I think this is why the data from AIDA and NIPI are not in completely different ball parks in the first place.

line 572: "two diatom species" does not sound right. This gives the impression that the diatoms themselves were aerosolized and examined, while you rather looked at the whole algal cultures.

lines 577-579: It might be better to split this in two sentences: "Our three main objectives were: first the comparison of the ice nucleating ability of two common phytoplankton species with Arctic microlayer samples, second examining the impact of the aerosolization technique on the results, and third deriving the sample variability over the entire mixed-phase cloud temperature range. Concerning these objectives, we can draw the following conclusions:"

line 581: "among" might be better than "within".

Fig. 7: Why is the freezing point depression not accounted for the AIDA data? In case this could not be made, it would make more sense to not do this correction at all.

line 584: Please exchange "triggered" by another term. The sentence is not clear to me, the way it is formulated now.

line 636: Above (lines 629-630), you mention "heat treatment test (not shown) . . . only a weak heat sensitivity". Here you say they are heat sensitive!?!? Check / revise!

Literature:

Creamean, J. M., J. N. Cross, R. Pickart, L. McRaven, P. Lin, A. Pacini, R. Hanlon, D. G. Schmale, J. Ceniceros, T. Aydell, N. Colombi, E. Bolger, and P. J. DeMott (2019), Ice Nucleating Particles Carried From Below a Phytoplankton Bloom to the Arctic Atmosphere, Geophys. Res. Lett., 46(14), 8572-8581, doi:10.1029/2019gl083039.

Gong, X., H. Wex, M. van Pinxteren, N. Triesch, K. W. Fomba, J. Lubitz, C. Stolle, B. Robinson, T. Müller, H. Herrmann, and F. Stratmann (2020), Characterization of aerosol particles at Cape Verde close to sea and cloud level heights - Part 2: ice nucleating particles in air, cloud and seawater, Atmos. Chem. Phys., 20, 1451-1468, doi:10.5194/acp-20-1451-2020.

Hoose, C., and O. Moehler (2012), Heterogeneous ice nucleation on atmospheric aerosols: A review of results from laboratory experiments, Atmos. Chem. Phys., 12, 9817–9854, doi:10.5194/acp-12-9817-2012.

Kanji, Z. A., L. A. Ladino, H. Wex, Y. Boose, M. Kohn, D. Cziczo, and M. Krämer (2017), Chapter 1: Overview of Ice Nucleating Particles, in Ice Formation and Evolution in Clouds and Precipitation: Measurement and Modeling Challenges, edited, Meteor. Monogr., doi:10.1175/AMSMONOGRAPHS-D-16-0006.1.

Ladino, L. A., G. B. Raga, H. Alvarez-Ospina, M. A. Andino-Enríquez, I. Rosas, L. Martínez, E. Salinas, J. Miranda, Z. Ramírez-Díaz, B. Figueroa, C. Chou, A. K. Bertram, E. T. Quintana, L. A. Maldonado, A. García-Reynoso, M. Si, and V. E. Irish

(2019), Ice-nucleating particles in a coastal tropical site, Atmos. Chem. Phys., 19(9), 6147-6165, doi:10.5194/acp-19-6147-2019.

McCluskey, C. S., T. C. J. Hill, R. S. Humphries, A. M. Rauker, S. Moreau, P. G. Strutton, S. D. Chambers, A. G. Williams, I. McRobert, J. Ward, M. D. Keywood, J. Harnwell, W. Ponsonby, Z. M. Loh, P. B. Krummel, A. Protat, S. M. Kreidenweis, and P. J. DeMott (2018), Observations of Ice Nucleating Particles Over Southern Ocean Waters, Geophys. Res. Lett., 45(21), 11989-11997, doi:10.1029/2018gl079981.

McCluskey, C. S., J. Ovadnevaite, M. Rinaldi, J. Atkinson, F. Belosi, D. Ceburnis, S. Marullo, T. C. J. Hill, U. Lohmann, Z. A. Kanji, C. O'Dowd, S. M. Kreidenweis, and P. J. DeMott (2018), Marine and Terrestrial Organic Ice-Nucleating Particles in Pristine Marine to Continentally Influenced Northeast Atlantic Air Masses, J. Geophys. Res.-Atmos., 123(11), 6196-6212, doi:10.1029/2017jd028033.

Murray, B. J., D. O'Sullivan, J. D. Atkinson, and M. E. Webb (2012), Ice nucleation by particles immersed in supercooled cloud droplets, Chem. Soc. Rev., 41, 6519-6554.

Polen, M., E. Lawlis, and R. C. Sullivan (2016), The unstable ice nucleation properties of Snomax (R) bacterial particles, J. Geophys. Res.-Atmos., 121(19), 11666-11678, doi:10.1002/2016jd025251.

Szyrmer, W., and I. Zawadzki (1997), Biogenic and anthropogenic sources of ice-forming nuclei: A review, BAMS, 78(2), 209-228.

---

## Referee Comment (RC2) · Anonymous Referee #2 · 29 May 2020

In their manuscript titled "Arctic marine ice nucleating aerosol: a laboratory study of microlayer samples and algal cultures", Luisa et al. describe findings from a series of ice nucleation measurements performed on sea surface microlayer (SML) samples collected from previous Arctic field campaigns and two culture phytoplankton species. This research topic is of current interest for the aerosol-cloud interaction community, particularly for remote regions and high latitudes. The introduction motivates the study and the descriptions of the approach and methods used in this study are detailed and well written, which is greatly appreciate. I have only one major concern, which relates to Section 3.3 (see general comment #5) and some specific minor comments. Overall, the manuscript is well written and these results do advance current knowledge related

to ice nucleating material in the marine environment. I recommend this manuscript for publication once these comments have been adequately addressed.

General Comments:

1. The title of the manuscript is pretty misleading – there are no measurements of Arctic marine ice nucleating particles, meaning these measurements were not made for aerosol collected in Arctic. I understand that the results may potentially have implications for the Arctic, but I recommend the authors consider changing the title to be more transparent about what this study entails.

2. Throughout the manuscript, it is difficult to know exactly what type of sample is being discussed: a bulk SML/culture sample, an nebulized aerosol sample, or a AEGOR aerosol sample. For example, it is not clear if Section 3.1 includes any measurements of aerosol samples or if it is strictly SML/culture samples. The section title is not very specific and Figure 1 has arrows pointed toward the "Droplet freezing experiments" picture from both the bulk sample and from the AEGOR, but I do not think there are measurements of AEGOR aerosol with the NIPI technique.

3. There are many instances where discussion on the interface between bulk seawater, SML, and aerosol is relevant for understanding the findings. The size-dependent aerosol composition is also relevant for interpreting the results from the two aerosol methods. I encourage the authors to consider a paragraph in the introduction that includes some of the literature on this topic and why studies such as this one are useful to address this knowledge gap in the context of ice nucleation research.

4. In calculating the ice nucleation site density, it's important to be clear and specific as to how the nebulizer will bias the ice nucleation site densities to higher values. Specifically, that the narrow size distribution with small particles that are likely more enriched in organic material compared to larger particles sizes will bias estimated ice nucleation site densities to higher values compared to natural aerosol and the AEGOR emissions.

5. My understanding is that Section 3.3. "Combined temperature regime – full ice nucleation spectra" aims to quantify the full ice nucleation temperature spectra from the different instruments, which includes measurements of both aerosol and bulk water/SML samples. To do so, the authors have estimated ice nucleation active site densities per mass of sea salt. I absolutely understand the experimental limitations that motivate this and I also think the authors include a thorough explanation as to why this approach may not be appropriate, which is very appreciated. This approach assumes that the ratio of the ice nucleation material to salt is equivalent in the bulk SML/culture samples and the aerosol. As the authors are aware, the transfer of organic material (likely responsible for the ice nucleation behavior) between the bulk water, SML, and aerosol phases is complex and varies depending on the solubility and surface active properties of the ice nucleation material. As such, I think presenting this analysis as "combined full ice nucleation spectra" is highly misleading to readers who may try to do the same in their own experiment without the careful consideration of discrepancies between bulk and aerosol composition or who may try to reuse the data. I do think these results are interesting and address the puzzling process of evaluating ice nucleation associated with the marine system (bulk, SML, and aerosol). Instead of presenting Section 3.3 as "full ice nucleation spectra", I suggest the authors to consider reframing these results as an approach for investigating the transfer of ice nucleation material from the bulk phase to the aerosol phase. Most of this discussion is already included, so it would require renaming the section and changing the order of text and therefore isn't really a major change.

Specific Minor Comments:

L15 – "we applied several aerosolisation techniques" – should this say "two aerosolisation techniques"?

L41 – the references listed for sea spray aerosol as an important INP source in remote regions includes only numerical modeling studies. This should be specified as such. There are also additional observational studies in remote regions that are cited in the

manuscript elsewhere and would also support this statement.

L57 –Why are these specific temperatures listed in reference to the DeMott et al. (2016) study? DeMott et al. (2016) evaluated INPs at a range of temperatures for their study -15 to -34 deg C for laboratory studies; -6 to -27 for ambient aerosol measurements (see Figure 1 from that manuscript).

L75 – "... suggested that absolute cell concentrations..." – is this referring to cell concentrations in air or in seawater?

L87 – "the ice nucleating potential of the aerosolised organic matter has not been examined in detail" – Do you specifically mean marine organic matter in the Arctic? Please be specific, as previous studies have investigated INPs associated with marine organic material and organic material in other settings.

Table 1 – Is this table necessary? Only a couple studies are mentioned in the introduction and several studies are missing if this is intended to be a full summary of marine INP studies. If you really want to include a table like this, I suggest including only the studies relevant for the Arctic region or laboratory studies since those are the focus of this paper. If it is decided that the authors want to include a table of all studies that have targeted ice nucleation observations of marine aerosol/SML/seawater, please take some time to be inclusive to all marine INP studies.

L112 – what is meant by "ex situ" ?

Table 3 – Are these for the bulk water samples or aerosol samples?

Table 3 - "we give in brackets how many mL of sample" – I think this should say parentheses, not brackets

L162 – Here, the order of the text suggests that these subsamples were with the AC-CACIA campaign, but the table lists them as ASCOS. Is the sentence stating "The surface microlayer water was collected from ...." out of order?

L178 – What does STN mean?

L190 – throughout the methods section, when referring to "samples", the authors should be clear if they are talking about the bulk samples/mixtures or aerosol samples. Here, I think "the cell concentrations of algae in the experiment" is referring to bulk samples/mixtures (not aerosol), but this is not clear here nor in the Table 3 description (see general comment 2).

L200 – Why were the tank water temperatures changed for the different experiments? Was this intentional? Studies have demonstrated that aerosol production is sensitive to temperature (e.g., Zábori et al., 2012), so curious if there was a reason and if the authors can elaborate on this detail.

Zábori, J., MatisÄĄns, M., Krejci, R., Nilsson, E. D., & Ström, J. (2012). Artificial primary marine aerosol production: a laboratory study with varying water temperature, salinity, and succinic acid concentration. Atmospheric Chemistry and Physics, 12(22), 10709–10724. https://doi.org/10.5194/acp-12-10709-2012

L210 – This last statement suggests that you can account for differences between the two aerosol generation techniques just by applying a dilution factor. However, an important difference between the nebulizer and the plunging jet is the size distribution, which is shown in Figure 2, and the corresponding organic composition of the generated aerosol because of the size-dependent composition of nascent sea spray aerosol. This was demonstrated in numerous studies, such as O'Dowd et al., 2004 (field evidence) and Prather et al., 2013 (laboratory evidence).

O'Dowd, C. D., Facchini, M. C., Cavalli, F., Ceburnis, D., Mircea, M., Decesari, S., et al. (2004). Biogenically driven organic contribution to marine aerosol. Nature, 431(7009), 676–680. https://doi.org/10.1038/nature02959

Prather, K. A., Bertram, T. H., Grassian, V. H., Deane, G. B., Stokes, M. D., DeMott, P. J., et al. (2013). Bringing the ocean into the laboratory to probe the chemical complex-

ity of sea spray aerosol. Proceedings of the National Academy of Sciences, 110(19), 7550–7555. https://doi.org/10.1073/pnas.1300262110

L234 – This is a large range of variability (180 to 900 nm) that spans an important size range for sea spray aerosol composition (see referenced in previous comment). Could the median diameters and widths be included in Table 2 to aid in interpreting the figures and data that follow?

L304 – Are all sizes of particles transmitted to the to the INKA instrument?

L321 – "sample under investigation" – are these samples bulk samples or aerosol collected onto filters? Based on Figure 1, it looks like the cold stage technique is applied to the bulk sample and the AEGOR samples, but there are no details describing how the aerosol are collected from the AEGOR and then analyzed with the NIPI method.

Figure 3b – it is very difficult to see the difference between the SM100C, SM100d, and SM100d nebulized marker colors. Also, is SM100b missing?

Figure 3 – Is the artificial seawater missing from this? It is listed in Table 2 as having been analyzed with the NIPI method.

L411 – Another possible important difference in the ASCOS high mol. w. sample is the aerosol sizes, which were mentioned previously as the smallest sizes observed from the nebulizer method. The surface area normalizes the data, but it should also be mentioned that the composition (i.e., possible ice nucleating material) is strongly size dependent for sea spray aerosol. Thus, the ASCOS high mol. w. aerosol sample may include smaller particles with greater organic mass fractions compared to the other aerosol samples, which further supports your finding (see general comment 4).

Figure 5 – Are all of these data for aerosol generated from the nebulizer or the AEGOR? There is a triangle in the legend, but I only see one set of data points plotted with triangles and Table2 includes 5 samples and the artificial seawater that were aerosolized with the AEGOR.

Figure 5 - By including the full range from the DeMott study (including uncertainties), the impression provided by this figure is that ns for marine aerosol spans 3 orders of magnitude and dessert dust is perfectly known. I think this is a bit misleading and I suggest the authors may want to consider using the marine INP ns parameterization from McCluskey et al. (2018) or the parameterizations used in Huang et al., 2018.

McCluskey, C. S., Ovadnevaite, J., Rinaldi, M., Atkinson, J., Belosi, F., Ceburnis, D., et al. (2018). Marine and Terrestrial Organic Ice-Nucleating Particles in Pristine Marine to Continentally Influenced Northeast Atlantic Air Masses. Journal of Geophysical Research: Atmospheres, 123(11), 6196–6212. https://doi.org/10.1029/2017JD028033

Huang, W. T. K., Ickes, L., Tegen, I., Rinaldi, M., Ceburnis, D., & Lohmann, U. (2018). Global relevance of marine organic aerosol as ice nucleating particles. Atmospheric Chemistry and Physics, 18(15), 11423–11445. https://doi.org/10.5194/acp-18-11423-2018

L458 – The comparison between AIDA and IKNA data is interesting. I have no issues with what is included in this discussion, but wonder if the authors could comment on additional impacts associated specifically with the unique particle composition. That is, once the solution droplet/particle effloresces and re-deliquesces, will it have the same ice nucleation activity as the particle that enters the AIDA chamber? Additionally, what do these results suggest for naturally occurring aerosol-cloud interactions and which (AIDA or INKA) is more representative of natural sea spray aerosol production, transport, activation and nucleation?

Section 3.3 – See General Comment 5.

L552 – what is "mist"?

L574 – I do not think "direct comparison" is an appropriate description for the analysis completed here because the ice nucleation ability of aerosol and bulk samples are not directly comparable. I suggest that the authors change this language to "We have

normalized all of the measurements by the salt mass present in the bulk and aerosol samples to investigate the ability of ice nucleating material to transfer to the aerosol phase" or similar (see General Comment 5).

L611 – "We also tentatively show that nebulisation enhances the ice nucleating ability of some cell cultures. We suggest that the aerosolisation process might rupture individual cells allowing ice nucleating macro-molecules to be dispersed through the aerosol population" – Please specify that this result only has implications for laboratory studies, not reality, because the nebulizer is not a naturally occurring phenomena at the ocean surface.

L623 –This is a great discussion. While it should not and is not be expected that the authors know every paper on this topic, I do want to point out two others that are extremtely relevant to this discussion: McCluskey et al., 2018 identified two marine INP types during mesocosm experiments and also include a discussion on the timing/conditions of their emissions and a very recent paper by Wilbourn et al., 2020 describe additional phytoplankton species and may be interesting to include.

McCluskey, C. S., Hill, T. C. J., Sultana, C. M., Laskina, O., Trueblood, J., Santander, M. V., Beall, C. M., Michaud, J. M., Kreidenweis, S. M., Prather, K. A., Grassian, V., and DeMott, P. J.: A Mesocosm Double Feature: Insights into the Chemical Makeup of Marine Ice Nucleating Particles, J. Atmos. Sci., 75, 2405-2423, 10.1175/JAS-D-17-0155.1, 2018

Wilbourn, E. K., Thornton, D. C. O., Ott, C., Graff, J., Quinn, P. K., Bates, T. S., et al. (2020). Ice Nucleation by Marine Aerosols Over the North Atlantic Ocean in Late Spring. Journal of Geophysical Research: Atmospheres, 125(4). https://doi.org/10.1029/2019JD030913

L629 – The heat tests would be interesting to see, especially for inferring ice nucleation material type/properties and since it is later mentioned that "The fact that marine INP are very small and heat sensitive" – is this a fact based on the data (not shown) or a

hypothesis based on previous measurements?

---

## Author Comment (AC1) · 16 Jul 2020

**Anonymous Referee #1: Review of "Arctic marine ice nucleating aerosol: a laboratory study of microlayer samples and algal cultures" by Ickes et al., submitted to ACPD**

The manuscript is a very well written and quite exhaustive study about the ice nucleation ability of different samples related to Arctic sea water (algal cultures and sea surface microlayer samples), examined with different measurement approaches. Measurement approaches include the examination of liquid samples on a cold-stage, and measuring aerosolized samples with the expansion chamber AIDA and the in-situ instrument INKA (a continuous flow diffusion chamber). For aerosol generation, two different methods were deployed, aerosolization and a plunging jet sea spray chamber.

Measurements were all done with great care and with up to date knowledge. The work describes and summarizes all measurements well, observed results are interpreted broadly and with great care. I particularly liked section 3.3, in which all different methods are compared, based on a normalization wrt. the salt content of the samples (and again with sufficient care, in that caveats of using this normalization are mentioned as well).

*We thank anonymous reviewer #1 for the positive review and the detailed comments on the manuscript. We have revised the manuscript accordingly (see track-changes in the manuscript). Our replies to your comments are given below in blue after the specific comment.*

Having said all these positive things, I did miss some final information about what was learned for the atmosphere from this study. It is clear that the contribution of "real world sea spray" to atmospheric aerosol and particularly to the INP fraction is a different topic. But the authors decided to submit their work to ACPD, and in that context it will benefit from some summarizing remarks about the applicability of results from this study for atmospheric implications.

*In our study we analyse the ice nucleating potential of Arctic sea spray aerosol, which could have an influence on Arctic mixed-phase clouds and thus the climate. We will further emphasise the atmospheric relevance of our results in the abstract and conclusions of our paper by adding the following sentences: "In the Arctic, these INPs can influence water-ice partitioning in low-level clouds and thereby the cloud lifetime, with consequences for the surface energy budget, sea ice formation and melt, and climate." (abstract); "We discuss our results in the context of aerosol-cloud interactions in the Arctic with a focus on furthering our understanding of which INP types may be important in the Arctic atmosphere." (abstract) and "From our study it is difficult to answer the question whether Arctic regions may have local marine sources of INPs and how much they influence Arctic mixed-phase clouds. At temperature above 248 K the ice nucleation activity of the investigated samples was very diverse, with some samples reaching a quite high median freezing temperature of 262 K, thus potentially being able to trigger freezing in Arctic mixed-phase clouds. The measurements in the temperature regime below 248 K on the other hand did not show that the samples were particularly ice active, especially when compared to dust, despite the fact that the results show an upper limit for $n_s$. Both measurements differentiated in the way the samples were analysed (bulk vs. aerosol phase). This was most relevant for the cultured samples, giving some hint that aerosolisation of cell cultures may change the ice nucleation activity of these, a process that could be important in the environment as well." (conclusions).*

Additionally, although I have no big concerns about this work, there is a rather longish number of remarks I give below. They are all not essential, but should be dealt with to make this already good work such that it then can be published in ACP.

**Specific remarks:**

line 37: "The types of aerosol particles that constitute good INP are uncertain (DeMott et al., 2010)." As the field of atmospheric ice nucleation research had a VERY strong revival in the last decade, this sentence is a) not really correct, and b) the citation is quite old. Since then, there were several review papers, and there is already an older one (Szyrmer & Zawadzki, 1997; Hoose & Möhler, 2012; Murray et al., 2012; Kanji et al., 2017). I don't want you to cite all of them, it's just to show you why I think this sentence needs revision (or deletion).

*There is still a lot of uncertainty around which aerosol particles make good INP and why. We changed the sentence to: Despite increasing interest in INP (Szyrmer and Zawadzki, 1997; Hoose and Möhler, 2012; DeMott et al., 2010), it is still uncertain which types of aerosol particles that constitute good INP in the atmosphere (Kanji et al., 2017).*

line 42: "sea spray aerosol could be an important source of INP" – again, it can't be expected that you give a complete review here, but as this is a focus of your work, I wanted to point out these papers on the topic: McCluskey et al., (2018a, b) and Creamean et al. (2019).

*We updated the references as recommended.*

line 48: "three main groups" - There is actually a quite new paper in which SML and airborne concentrations were connected (including cloud water): Gong et al. (2020). For air masses that were continentally and/or mineral dust influenced (Cape Verde), INP concentrations in the SML did not explain atmospheric INP concentrations - that does not say anything about the remote oceans, but is a piece in the puzzle, nevertheless, which is worth mentioning, as this helps closing a gap between ocean and atmosphere.

*Thanks for bringing our attention to this paper, we added it to Table 1 and also added it in the introduction.*

lines 84/85 and line 95-97: You write: "Another goal of this study was to improve our understanding of whether Arctic marine regions may have local sources of marine INPs." and "Through comparison of the ice nucleation activity of artificial seawater containing Melosira arctica with that of the SML samples we aim to shed light on how representative relevant algal cultures are for Arctic marine INP." - Comments on that in the summary (that I proposed above) would be highly welcome, although it is clear that it is not straightforward to draw conclusions on atmospheric concentrations of sea spray aerosol from artificial lab work. But there are results you can summarize!

*As stated above, we added a few more statements on that in the conclusions.*

Table 1: As you try to give an overview here, including the papers I referred to above makes sense (McCluskey et al., 2018a,b; Creamean et al., 2019; Gong et al., 2020).And there is one more for coastal Mexico by Ladino et al., (2019) which may fit.

*We added the mentioned papers to the table (and a couple of other ones as well).*

line 323: "sample is well mixed, so that particles are distributed uniformly, and each droplet is representative." Actually, if the INP concentrations are so high that this is true,then each well freezes at the same time, yielding a super-steep freezing curve. When the sample is then diluted or is already more diluted to begin with (so that conditions are those for which measurements typically are made), a less steep increase of FF with decreasing temperature is observed. But then, strictly speaking, INP are Poisson-distributed and this here does not apply any more. Therefore, you could say "... so that particles are distributed randomly."

*We changed it to: "...sample is well mixed, so that particles are distributed randomly, and each droplet is representative of the sample as a whole, meaning each one has an approximately equal probability of containing an INP active at a given temperature."*

lines 385-389: Due to the dilution that was done, comparing FF does not make sense, and the figure could have only been shown for INP concentrations (normalized) – at least a FF figure cannot be interpreted in this way. Please revise.

*We removed the FF plot and changed the text accordingly.*

lines 408/409: This sentence here gives a wrong impression. It becomes clear in the next chapter (3.3), that there really is a lower variability for the AIDA data at the lower temperatures. But before normalizing data from the different instruments (as you do, based on the sea salt concentration below), you should refrain from any kind of comparison and discussion thereof.

*We agree that this comparison here is a bit far fetched since we compare the non-normalised results, which one probably should not do. We changed the sentence to: "The various SML samples show little variation at temperatures below 248 K when probed in the AIDA chamber, meaning that the SML samples all exhibited similar ice nucleation activity ($n_s$ of $10^9$ $m^{-2}$ at temperatures between 240 - 244 K) and the individual $n_s(T)$-curves of the AIDA measurements form a rather compact block of data (Fig. 5)."*

lines 441-442: This conclusion confuses me a little. You used the dry(!) particle number size distribution to derive the surface area for both particle generation methods when you normalized. This basically means that the contribution of different aerosol particle types (such as particles consisting purely of salt, or of having a mix between salt and organics, ...) to the overall aerosol was similar, independent of the particle generation. And therefore, this conclusion does not hold.

*The dry particle number size distribution consists of all particles (pure sea salt, mixtures of salt and organics, ...) and there is no discrimination based on particle composition. Thus we cannot derive*

*any information on composition from the size distributions. However, we can conclude from the measurements of the ice nucleation activity (AEGOR results vs. SML results) that the aerosol composition (organics/salt) has changed.*

line 449: And of course, for atmospheric relevancy, also the abundance of these different particle types has to be accounted for. Mentioning this here would be good.

*We added a sentence on the abundance of mineral dust vs. sea spray here: "In the (High) Arctic both transported dust and sea spray aerosol (transported or locally originated) can be present (see Willis et al. 2018 for a thorough review of literature). However, which source is dominant for ice nucleation might be locally very different. In regions dominated by sea spray aerosol the fraction of organic matter within the aerosol population is another uncertainty.".*

lines 535-536: "the diluted sample having higher nm values compared to the undiluted sample in the same temperature regime." This may be an indication that the back-ground was hit, i.e., one already measures background, dilutes more and again only measured background, but normalizes to a higher dilution. Check if this is the case here, and if yes, omit the data.

*We removed the sentence since it was misleading. We do not think it is the background influence in this case but rather an effect of run-to-run uncertainty. We have taken backgrounds into account when analysing the data.*

lines 582-583: To be able to draw this conclusion, results would have to be normalized to "atmospheric algal content", which wasn't done and (as you argue above) is difficult, even amongst the NIPI data. This should be mentioned.

*We changed the sentence to: "As the investigated algae species show less ice activity in the temperature regime above 248 K compared to the natural field samples, we conclude that they, especially Melosira arctica, cannot explain the freezing at the high temperatures.". And added: "A normalisation of the samples to the atmospheric algal content would be needed to quantify this observation.".*

**Technical comments:**

The word "freezing depression" is used consistently. "freezing point depression" is the more correct term, right? Same for "aerosolisation" -¿ "aerosolization"?

*Thanks for pointing that out, we changed "freezing depression" consistently to "freezing point depression".*
*We use British English, where "aerosolisation" is the correct term.*

Figure 1: When initially looking at this figure, I wondered about the meaning of the arrow on the left (in the middle), going from the plunging jet tube to the droplet freezing experiment. It became clear later on. It's probably a matter of taste to leave it here or to delete it – I just wanted to point out my initial confusion.

*Thanks for pointing that out, we simplified the figure.*

lines 137-139: Doesn't the nutrient content determine the growth rate? - It puzzled me that you can somehow set things such that you get high growth with low nutrient content (middle case). Or is the growth determined separately, and you just already give these observations here? Then that should be made clear.

*Nutrients, but also their stoichiometric ratios, determine growth. In treatment 1, we had algae that were both dividing fast and had a large cell size. In treatment 2, the cell division rate was high, but the cell size was small (= phosphorus limitation). In treatment 3, both parameters were low (= nitrogen AND phosphorus limitation). We added this explanation to the manuscript as well.*

line 215ff: Upon reading this for the first time, I was confused about the influence this low temperature in AIDA during the preparation phase would have on the measurements. This is nicely explained later, and it would be good to point out that an explanation on the reason for choosing such a low temperature will follow.

*We changed the sentence to: "In the AIDA chamber, typically held at 250 K and a relative humidity of 78% during aerosol injection (see Sect. 2.4 for more details and an explanation on the low temperature), the aerosol particles were suspended as supercooled aqueous solution droplets.".*

line 247: "The smaller...vessel" could better be introduced here as "A smaller 3.7m3-sized stainless steel vessel located in the vicinity of AIDA...".

*We changed this accordingly to: "A smaller 3.7 $m^3$-sized stainless steel vessel is located in the vicinity of AIDA. It is referred to as the APC (aerosol preparation and characterisation) chamber and can only be operated at ambient temperature.".*

line 306: "sample air flow is sheathed by dry particle free synthetic air" – that is only true initially - water vapor diffuses through the sheath air flows and the sample flow from one plate (the warmer) to the other. Consider reformulation.

*This is true, the sample air flow is sheathed by prior dried synthetic air. We changed to wording to: "sample air flow is sheathed by particle free synthetic air (initially dry)".*

line 371: Concerning a difference in ice nucleation temperature observed for different cooling rates, the papers you mention show that the influence of the cooling rate is rather small. Please add that explicitly! As you mention, this can rather not explain the differences you observe.

*We changed the sentence to: "The temperature at which 50% of the droplets are frozen has been shown to decrease with increased cooling rate in Wright and Petters (2013); Herbert et al. (2014), also the dependence was shown to be rather small.".*

line 377: Concerning the loss of ice activity during storage, there is a paper on that for Snomax (Polen et al., 2016).

*Thank you very much for this reference. We included it in the respective discussion.*

Figure 3: 1) The different shades of bluish green in b) are difficult to distinguish, engraved by the opacity changes. Maybe additionally also change the symbol styles between samples? Or use a broader range of colors? (Although it is nice how you use consistent colors throughout the manuscript for the separate samples.)

*We changed the blueish color of SM100c and hope it is better now (we did not want to add confusion by other symbols or change the consistency in between all plots).*

Figure 3: 2) Also, check the legend in panel b): SM 100d appears twice, SM 100b not at all.

*It is correct that the SM100d appears twice: one is the pure/original sample, the other one is the nebulised sample as indicated in the legend and by different symbols. We added SM100b to the plot and added some text/discussion on this sample in section 3.1.*

Figure 3: 3) "Two duplicate samples of SM100 (SM100a and SM100c)" - do you refer to the two bags in which the sample was delivered? In the text these are "a" and "b",while "c" is the one that was stored. Check and homogenize.

*SM100a and SM100c is coming from two different bags of the sample (bag 1: SM100a; bag 2: SM100b). SM100c is a subsample of SM100b (2. bag) after two months storage. To reduce the ambiguity we changed the text in the caption to: "Two duplicate samples of SM100 (SM100a and SM100b) are reflecting the variability of the sample. SM100a and SM100b are from two bags collected from the same culture. SM100c is a sub-sample of SM100b after 2 months storage. SM100d, a sub-sample of SM100b, was in storage for 10 months, and was then nebulised and retested to determine the effect of the aerosolisation on the sample."*

line 417: Delete "of" (at "contained of...gels") or replace "contained" by "consisted".

*Corrected.*

Fig. 5, 7, 8, and 9: Maybe use a separate legend explaining the different symbol types(AIDA, INKA, ...), as it was done for Fig. 6. - It's a bit confusing to see one entry in the legend saying "SMLx", and then another sample is "AIDA".

*Adapted.*

line 539: Add "is" between "this" and "related".

*Corrected.*

line 561: Shouldn't "since" rather be replaced by an "although"? If it's assumed that INP are small, and not intact cells, aerosolization might not do additional harm. In fact,I think this is why the data from AIDA and NIPI are not in completely different ball parks in the first place.

*"The whole section and sentence changed (due to other revisions) and now reads as: "The aerosolisation technique might exert more of an influence on the cultured samples compared to the microlayer samples, where the INP are thought to be associated with submicron organic detritus, rather than intact cells.".*

line 572: "two diatom species" does not sound right. This gives the impression that the diatoms themselves were aerosolized and examined, while you rather looked at the whole algal cultures.

*We changed the wording to: "two diatom cultures".*

lines 577-579: It might be better to split this in two sentences: "Our three main objectives were: first the comparison of the ice nucleating ability of two common phytoplankton species with Arctic microlayer

samples, second examining the impact of the aerosolization technique on the results, and third deriving the sample variability over the entire mixed-phase cloud temperature range. Concerning these objectives, we can draw the following conclusions:"

*Corrected.*

line 581: "among" might be better than "within".
*Corrected.*

Fig. 7: Why is the freezing point depression not accounted for the AIDA data? In case this could not be made, it would make more sense to not do this correction at all.
*The freezing point depression in AIDA is negligible (< 0.1 K) because the seed aerosol particles are activated to > 10 μm sized cloud droplets with a remaining solute concentration of < 0.1 wt%, that is, a water activity close to 1 and has therefore not been accounted for.*

line 584: Please exchange "triggered" by another term. The sentence is not clear to me, the way it is formulated now.
*The sentence was changed to: "This result indicates that the INPs active at the highest temperatures are not one of the two types of phytoplankton cells studied or their exudates.".*

line 636: Above (lines 629-630), you mention "heat treatment test (not shown)...only a weak heat sensitivity". Here you say they are heat sensitive!?!? Check / revise!
*Lines 629-630 refer to only one sample (SM100), which we did examine in terms of heat sensitivity. Line 636 refers to the general knowledge on marine INP - we added some citations now to emphasise this and rephrased it. We agree that this seems contradicting, but we did not want to derive too much from one single measurement. More measurements are needed for further conclusions.*

**Literature:**
Creamean, J. M., J. N. Cross, R. Pickart, L. McRaven, P. Lin, A. Pacini, R. Hanlon, D.G. Schmale, J. Ceniceros, T. Aydell, N. Colombi, E. Bolger, and P. J. DeMott (2019),Ice Nucleating Particles Carried From Below a Phytoplankton Bloom to the Arctic Atmosphere, Geophys. Res. Lett., 46(14), 8572-8581, doi:10.1029/2019gl083039.

Gong, X., H. Wex, M. van Pinxteren, N. Triesch, K. W. Fomba, J. Lubitz, C. Stolle,B. Robinson, T. Müller, H. Herrmann, and F. Stratmann (2020), Characterization of aerosol particles at Cape Verde close to sea and cloud level heights - Part 2: ice nucleating particles in air, cloud and seawater, Atmos. Chem. Phys., 20, 1451-1468,doi:10.5194/acp-20-1451-2020.

Hoose, C., and O. Moehler (2012), Heterogeneous ice nucleation on atmospheric aerosols: A review of results from laboratory experiments, Atmos. Chem. Phys., 12,9817–9854, doi:10.5194/acp-12-9817-2012.

Kanji, Z. A., L. A. Ladino, H. Wex, Y. Boose, M. Kohn, D. Cziczo, and M. Krämer(2017), Chapter 1: Overview of Ice Nucleating Particles, in Ice Formation and Evolution in Clouds and Precipitation: Measurement and Modeling Challenges, edited, Meteor. Monogr., doi:10.1175/AMSMONOGRAPHS-D-16-0006.1.

Ladino, L. A., G. B. Raga, H. Alvarez-Ospina, M. A. Andino-Enríquez, I. Rosas,L. Martínez, E. Salinas, J. Miranda, Z. Ramírez-Díaz, B. Figueroa, C. Chou, A. K.Bertram, E. T. Quintana, L. A. Maldonado, A. García-Reynoso, M. Si, and V. E. Irish, Ice-nucleating particles in a coastal tropical site, Atmos. Chem. Phys., 19(9),6147-6165, doi:10.5194/acp-19-6147-2019.

McCluskey, C. S., T. C. J. Hill, R. S. Humphries, A. M. Rauker, S. Moreau, P. G. Strut-ton, S. D. Chambers, A. G. Williams, I. McRobert, J. Ward, M. D. Keywood, J. Harnwell,W. Ponsonby, Z. M. Loh, P. B. Krummel, A. Protat, S. M. Kreidenweis, and P. J. DeMott(2018), Observations of Ice Nucleating Particles Over Southern Ocean Waters, Geo-phys. Res. Lett., 45(21), 11989-11997, doi:10.1029/2018gl079981.

McCluskey, C. S., J. Ovadnevaite, M. Rinaldi, J. Atkinson, F. Belosi, D. Ceburnis, S.Marullo, T. C. J. Hill, U. Lohmann, Z. A. Kanji, C. O'Dowd, S. M. Kreidenweis, and P.J. DeMott (2018), Marine and Terrestrial Organic Ice-Nucleating Particles in Pristine Marine to Continentally Influenced Northeast Atlantic Air Masses, J. Geophys. Res.-Atmos., 123(11), 6196-6212, doi:10.1029/ 2017jd028033.

Murray, B. J., D. O'Sullivan, J. D. Atkinson, and M. E. Webb (2012), Ice nucleation byparticles immersed in supercooled cloud droplets, Chem. Soc. Rev., 41, 6519-6554.

Polen, M., E. Lawlis, and R. C. Sullivan (2016), The unstable ice nucleation properties of Snomax (R) bacterial particles, J. Geophys. Res.-Atmos., 121(19), 11666-11678,doi:10.1002/2016jd025251.

Szyrmer, W., and I. Zawadzki (1997), Biogenic and anthropogenic sources of ice-forming nuclei: A review, BAMS, 78(2), 209-228.

**References**

DeMott, P. J., Prenni, A. J., Liu, X., Kreidenweis, S. M., Petters, M. D., Twohy, C. H., Richardson, M. S., Eidhammer, T., and Rogers, D. C.: Predicting global atmospheric ice nuclei distributions and their impacts on climate, Proceedings of the National Acadamy of Science of the United States of America (PNAS), 107, 11 217–11 222, 2010.

Hoose, C. and Möhler, O.: Heterogeneous ice nucleation on atmospheric aerosols: a review of results from laboratory experiments, Atmos. Chem. Phys., 12, 9817–9854, https://doi.org/10.5194/acp-12-9817-2012, 2012.

Kanji, Z. A., Ladino, L. A., Wex, H., Boose, Y., Burkert-Kohn, M., Cziczo, D. J., and Krämer, M.: Overview of Ice Nucleating Particles, Meteorological Monographs, 58, 1.1–1.33, https://doi.org/10.1175/AMSMONOGRAPHS-D-16-0006.1, 2017.

Szyrmer, W. and Zawadzki, I.: Biogenic and anthropogenic sources of ice-forming nuclei: A review, Bulletin of the American Meteorological Society, 78, 209–228, 1997.

---

## Author Comment (AC2) · 16 Jul 2020

**Anonymous Referee #2;**

In their manuscript titled "Arctic marine ice nucleating aerosol: a laboratory study of microlayer samples and algal cultures", Luisa et al. describe findings from a series of ice nucleation measurements performed on sea surface microlayer (SML) samples collected from previous Arctic field campaigns and two culture phytoplankton species.This research topic is of current interest for the aerosol-cloud interaction community, particularly for remote regions and high latitudes. The introduction motivates the study and the descriptions of the approach and methods used in this study are detailed and well written, which is greatly appreciate. I have only one major concern, which relates to Section 3.3 (see general comment #5) and some specific minor comments. Overall,the manuscript is well written and these results do advance current knowledge related to ice nucleating material in the marine environment.

I recommend this manuscript for publication once these comments have been adequately addressed.

*We thank anonymous reviewer #2 for the positive review and the detailed comments on the manuscript. We have revised the manuscript accordingly (see track-changes in the manuscript). Our replies to your comments are given below in blue after the specific comment.*

**General Comments:**

1. The title of the manuscript is pretty misleading – there are no measurements of Arctic marine ice nucleating particles, meaning these measurements were not made for aerosol collected in Arctic. I understand that the results may potentially have implications for the Arctic, but I recommend the authors consider changing the title to be more transparent about what this study entails.

*We changed the title to: "The ice nucleating activity of Arctic sea surface microlayer samples and marine algal cultures".*

2. Throughout the manuscript, it is difficult to know exactly what type of sample is being discussed: a bulk SML/culture sample, an nebulized aerosol sample, or a AEGOR aerosol sample. For example, it is not clear if Section 3.1 includes any measurements of aerosol samples or if it is strictly SML/culture samples. The section title is not very specific and Figure 1 has arrows pointed toward the "Droplet freezing experiments" picture from both the bulk sample and from the AEGOR, but I do not think there are measurements of AEGOR aerosol with the NIPI technique.

*Section 3.1 is only bulk SML/culture samples. We made this clearer in the manuscript by adding it to the section title. We also adapted Fig. 1 to reduce the confusion and adjusted all the labels or captions in Figure 3 and 4.*

3. There are many instances where discussion on the interface between bulk seawater, SML, and aerosol is relevant for understanding the findings. The size-dependent aerosol composition is also relevant for interpreting the results from the two aerosol methods. I encourage the authors to consider a paragraph in the introduction that includes some of the literature on this topic and why studies such as this one are useful to address this knowledge gap in the context of ice nucleation research.

*Thanks for bringing this up. We added a paragraph in section 2.2 on what we expect from the two different aerosolisation methods and what we want to address with our study: "We expect that aerosolisation of the samples with the nebulizer results in an upper estimate of INP because the undiluted SML (or cultured) samples are aerosolised whereas AEGOR is aerosolising a dilution of the samples with artificial seawater, which could result in a lower estimate of INP. However, it is not only the dilution factor in the sea spray simulation chamber (see Table 3), which has to be accounted for. The aerosolisation process itself is different in AEGOR compared to the nebuliser. In the nebuliser the suspension is well mixed, while in AEGOR the aerosol particles are formed from an organic enriched surface microlayer at the top of the tank. That leads to different expectations depending on the sample type. For the SML samples we would not expect such a huge difference due to this aspect. Here, we aerosolise in one case the pure well mixed SML (nebuliser), while in the other case we aerosolise the SML that has formed in AEGOR, which should be similar to the original SML sample. For the cultured samples, however, we would expect a larger influence. In AEGOR the phytoplankton material is floating at the surface of the tank leading to organic enriched aerosol particles during the aerosolisation, while the nebuliser might produce less enriched aerosol particles due to the mixing of the sample. Note that this might depend on the algae culture as well. Another crucial aspect of the two different aerosolisation methods is the size distribution and the resulting chemical composition of the generated aerosol. It was demonstrated in the laboratory and as well measured in the field, that for sea spray aerosol the organic composition of the aerosol particles and the generated size distribution are related (O'Dowd, C. D. et al., 2004; Prather et al., 2013). One interesting aspect of our study is to see the influence of all the aspects mentioned above and to check if*

*the diluted samples aerosolised with AEGOR show a similar or a lower freezing signal compared to the aerosolised pure samples.".*

4. In calculating the ice nucleation site density, it's important to be clear and specific as to how the nebulizer will bias the ice nucleation site densities to higher values. Specifically, that the narrow size distribution with small particles that are likely more enriched in organic material compared to larger particles sizes will bias estimated ice nucleation site densities to higher values compared to natural aerosol and the AEGOR emissions.

*Both the nebulizer and AEGOR produce rather broad size distributions (see Fig. 2). There is therefore no reason to assume that the ice nucleation active site densities calculated from the nebulizer measurements are overestimated due to that. However, the nebulizer results are an upper estimate because the undiluted SML (or also cultured) samples are aerosolised whereas AEGOR is aerosolising a dilution of the samples with artificial seawater. One interesting aspect of our study was to check if the diluted samples aerosolised with AEGOR show a similar or a lower freezing signal compared to the aerosolised pure samples. We did measurements with both aerosolisation methods for five samples: SM100, SM10, MA100, SML5 and SML8. We did observe a similar freezing signal for SML5 and MA100 when aerosolised with AEGOR, despite the dilution. The measurements of SML8, SM100 and SM10 did not exhibit a detectable freezing signal above the background when aerosolised with AEGOR.*
*We added to section 3.2: "However, both the nebulizer and AEGOR are not producing very narrow size distributions (see Fig. 2).".*
*We also added some more discussion on the expected results from both aerosolisation techniques in section 2.2.*

5. My understanding is that Section 3.3. "Combined temperature regime – full ice nucleation spectra" aims to quantify the full ice nucleation temperature spectra from the different instruments, which includes measurements of both aerosol and bulk water/SML samples. To do so, the authors have estimated ice nucleation active site densities per mass of sea salt. I absolutely understand the experimental limitations that motivate this and I also think the authors include a thorough explanation as to why this approach may not be appropriate, which is very appreciated. This approach assumes that the ratio of the ice nucleation material to salt is equivalent in the bulk SML/culture samples and the aerosol. As the authors are aware, the transfer of organic material (likely responsible for the ice nucleation behavior) between the bulk water, SML, and aerosol phases is complex and varies depending on the solubility and surface active properties of the ice nucleation material. As such, I think presenting this analysis as "combined full ice nucleation spectra" is highly misleading to readers who may try to do the same in their own experiment without the careful consideration of discrepancies between bulk and aerosol composition or who may try to reuse the data. I do think these results are interesting and address the puzzling process of evaluating ice nucleation associated with the marine system (bulk, SML, and aerosol). Instead of presenting Section 3.3 as "full ice nucleation spectra", I suggest the authors to consider reframing these results as an approach for investigating the transfer of ice nucleation material from the bulk phase to the aerosol phase. Most of this discussion is already included,so it would require renaming the section and changing the order of text and therefore isn't really a major change.

*We agree with the reviewer and shifted the focus of this section on the bulk- vs. aerosol measurements. We also changed the title accordingly. Both measurements methods are widely used and in our study we could investigate if both techniques complement each other or differ when normalised and brought into one context.*
*The new section reads now:*

[revised manuscript text omitted]

**Specific Minor Comments:**

L15 – "we applied several aerosolisation techniques" – should this say "two aerosolisation techniques"?
*Yes, corrected.*

L41 – the references listed for sea spray aerosol as an important INP source in remote regions includes only numerical modeling studies. This should be specified as such.There are also additional observational studies in remote regions that are cited in the manuscript elsewhere and would also support this statement.
*We have added further literature to support our statement.*

L57 – Why are these specific temperatures listed in reference to the DeMott et al. (2016) study?

DeMott et al. (2016) evaluated INPs at a range of temperatures for their study -15 to -34 deg C for laboratory studies; -6 to -27 for ambient aerosol measurements (see Figure 1 from that manuscript).

*It refers to Fig. 2 in the DeMott et al. 2016 study, where the authors tried to relate the INP conc. at these temperatures to TOC or Chlorophyll a. However, we agree that this is confusing and removed the parantheses with the specific temperatures in the manuscript.*

L75 – "...suggested that absolute cell concentrations..." – is this referring to cell concentrations in air or in seawater?

*It is the cell concentrations of the phytoplankton - we added this in the manuscript. The sentence now reads: "It has been suggested that absolute cell concentrations of the phytoplankton are not the sole determining factor for aerosol flux and that aerosol size distribution can be affected by the growth conditions of the microorganisms.".*

L87 – "the ice nucleating potential of the aerosolised organic matter has not been examined in detail" – Do you specifically mean marine organic matter in the Arctic? Please be specific, as previous studies have investigated INPs associated with marine organic material and organic material in other settings.

*Yes, that refers to the Arctic as this is the focus of our study. We have modified this sentence so it now reads: "..., the ice nucleating potential of the aerosolised organic matter has not been examined in detail for the Arctic region.".*

Table 1 – Is this table necessary? Only a couple studies are mentioned in the introduction and several studies are missing if this is intended to be a full summary of marine INP studies. If you really want to include a table like this, I suggest including only the studies relevant for the Arctic region or laboratory studies since those are the focus of this paper. If it is decided that the authors want to include a table of all studies that have targeted ice nucleation observations of marine aerosol/SML/seawater,please take some time to be inclusive to all marine INP studies.

*We did add further studies to this table to give a complete literature overview on marine INP studies. Note that we only mention articles that explicitly discuss marine sources of INP and focus on the mixed-phase cloud temperature regime.*

L112 – what is meant by "ex situ" ?

*It means that the aerosol particles have been sampled from the AIDA chamber and investigated with INKA, but not within the AIDA chamber using the expansion cooling experiment. To avoid confusion we removed the "ex situ".*

Table 3 – Are these for the bulk water samples or aerosol samples?

*These are for the bulk water samples - we specified it in the table caption, which now reads: "Characteristics of the bulk samples used during the study...".*

Table 3 - "we give in brackets how many mL of sample" – I think this should say parentheses, not brackets

*Corrected.*

L162 – Here, the order of the text suggests that these subsamples were with the ACCACIA campaign, but the table lists them as ASCOS. Is the sentence stating "The surface microlayer water was collected from...." out of order?

*It is only the sampling catamaran which is the same as during the ACCACIA campaign. Otherwise it is clearly stated that this paragraph is about the ASCOS samples.*

L178 – What does STN mean?

*It stands for Station. This is not of relevance for the paper - we decided to keep the same labels as in the original papers but here they can just be seen as labels (the meaning does not matter).*

L190 – throughout the methods section, when referring to "samples", the authors should be clear if they are talking about the bulk samples/mixtures or aerosol samples. Here, I think "the cell concentrations of algae in the experiment" is referring to bulk samples/mixtures (not aerosol), but this is not clear here nor in the Table 3 description(see general comment 2).

*We adapted the text here: "The cell concentrations of algae in the AEGOR tank...".*

L200 – Why were the tank water temperatures changed for the different experiments? Was this intentional? Studies have demonstrated that aerosol production is sensitive to temperature (e.g., Zábori et al., 2012), so curious if there was a reason and if the authors can elaborate on this detail.

Zábori, J., MatisÄ Ans, M., Krejci, R., Nilsson, E. D., & Ström, J. (2012). Artificialprimary marine aerosol production: a laboratory study with varying water temperature,salinity, and succinic acid

concentration. Atmospheric Chemistry and Physics, 12(22),10709–10724. https://doi.org/10.5194/acp-12-10709-2012

*We chose the temperatures to be the same as the temperature that was used for growing the cultures. For the SML samples we did choose a realistic temperature for the sea surface in the Arctic.*

L210 – This last statement suggests that you can account for differences between the two aerosol generation techniques just by applying a dilution factor. However, an important difference between the nebulizer and the plunging jet is the size distribution, which is shown in Figure 2, and the corresponding organic composition of the generated aerosol because of the size-dependent composition of nascent sea spray aerosol. This was demonstrated in numerous studies, such as O'Dowd et al., 2004 (field evidence) and Prather et al., 2013 (laboratory evidence).

O'Dowd, C. D., Facchini, M. C., Cavalli, F., Ceburnis, D., Mircea, M., Decesari, S., et al.(2004). Biogenically driven organic contribution to marine aerosol. Nature, 431(7009),676–680. https://doi.org/10.1038/nature02959

Prather, K. A., Bertram, T. H., Grassian, V. H., Deane, G. B., Stokes, M. D., DeMott, P.J., et al. (2013). Bringing the ocean into the laboratory to probe the chemical complexity of sea spray aerosol. Proceedings of the National Academy of Sciences, 110(19),7550–7555. https://doi.org/10.1073/pnas.1300262110

*We agree that the sentence was a bit misleading formulated. We changed it to: "Given these differences, comparison of the ice activity of aerosol generated by these two techniques should enable us to determine whether INP material is preferentially aerosolised by bubble-bursting.". We also added an additional paragraph here emphasising more what the differences between the nebuliser and AEGOR are and what we expect for the different samples.*

L234 – This is a large range of variability (180 to 900 nm) that spans an important size range for sea spray aerosol composition (see referenced in previous comment). Could the median diameters and widths be included in Table 2 to aid in interpreting the figures and data that follow?

*We have added the median particle diameters as well as the width (geometric standard deviation) of the fitted size distribution for all the AIDA expansions to Table 2.*

L304 – Are all sizes of particles transmitted to the to the INKA instrument?

*We assume that the particle loss in the sampling line is negligible, as shown in DeMott et al., The Fifth International Workshop on Ice Nucleation phase 2 (FIN-02): laboratory intercomparison of ice nucleation measurements, AMT, 11, 11, 6231–6257, 2018.*

L321 – "sample under investigation" – are these samples bulk samples or aerosol collected onto filters? Based on Figure 1, it looks like the cold stage technique is applied to the bulk sample and the AEGOR samples, but there are no details describing how the aerosol are collected from the AEGOR and then analyzed with the NIPI method.

*We changed the sentence to: "To do so, the droplets of the sample under investigation (if not explicitly otherwise mentioned this is a bulk sample) are pipetted onto a silanised glass slide, which serves as a hydrophobic substrate.". We also added after the description of the different AEGOR samples for the µL-NIPI: "Note that all these samples are bulk samples." and changed the labels in Figure 4.*

Figure 3b – it is very difficult to see the difference between the SM100C, SM100d, and SM100d nebulized marker colors. Also, is SM100b missing?

*We adapted the color scheme since that was also criticised by reviewer #1. We also added sample SM100b and some discussion on this sample.*

Figure 3 – Is the artificial seawater missing from this? It is listed in Table 2 as having been analyzed with the NIPI method.

*We added it to Figure 3.*

L411 – Another possible important difference in the ASCOS high mol. w. sample is the aerosol sizes, which were mentioned previously as the smallest sizes observed from the nebulizer method. The surface area normalizes the data, but it should also be mentioned that the composition (i.e., possible ice nucleating material) is strongly size dependent for sea spray aerosol. Thus, the ASCOS high mol. w. aerosol sample may include smaller particles with greater organic mass fractions compared to the other aerosol samples, which further supports your finding (see general comment 4).

*That could be an additional explanation for the difference in the AIDA measurements. We added: "The size distribution of the nebulised ASCOS high-molecular weight sample was the smallest compared to the other samples, which might have an influence on the ice nucleation activity as well since the*

*chemical composition of sea spray aerosol is highly size dependent. This sample might consist of smaller particles with a larger organic mass fraction compared to the other samples". However, this sample was independent of the size distribution more concentrated in organic mass due to the treatment. Note, that this sample was also the most effective in the µL-NIPI measurement.*

Figure 5 – Are all of these data for aerosol generated from the nebulizer or the AEGOR? There is a triangle in the legend, but I only see one set of data points plotted with triangles and Table 2 includes 5 samples and the artificial seawater that were aerosolized with the AEGOR.

*There are five samples which were aerosolised with AEGOR (SM100, SM10, MA100, SML5, SML8) as mentioned in Table 2. Figure 5 shows two of these samples (SML5 and SML8), while Figure 6 shows the other three (SM100, SM10, MA100). For most of the samples the freezing signal from the AEGOR measurement was below the background, so that there are no datapoints to be shown. That is for SML8, SM100 and SM10, meaning that Figure 5 and 6 indeed each only includes one AEGOR measurement (SML5 in Figure 5 and MA100 in Figure 6). This is mentioned in the text at the beginning of section 3.2 ("With respect to the experiments where AEGOR was used for aerosol generation, some samples did not exhibit a detectable freezing signal above the background (SM100, SM10, and SML8) and are therefore not included.").*

Figure 5 - By including the full range from the DeMott study (including uncertainties), the impression provided by this figure is that ns for marine aerosol spans 3 orders of magnitude and dessert dust is perfectly known. I think this is a bit misleading and I suggest the authors may want to consider using the marine INP ns parameterization from McCluskey et al. (2018) or the parameterizations used in Huang et al., 2018.

McCluskey, C. S., Ovadnevaite, J., Rinaldi, M., Atkinson, J., Belosi, F., Ceburnis, D.,et al. (2018). Marine and Terrestrial Organic Ice-Nucleating Particles in Pristine Marine to Continentally Influenced Northeast Atlantic Air Masses. Journal of Geophysical Research: Atmospheres, 123(11), 6196–6212. https://doi.org/10.1029/2017JD028033

Huang, W. T. K., Ickes, L., Tegen, I., Rinaldi, M., Ceburnis, D., & Lohmann, U. (2018).Global relevance of marine organic aerosol as ice nucleating particles. Atmospheric Chemistry and Physics, 18(15), 11423–11445. https://doi.org/ 10.5194/acp-18-11423-2018

*We added the McCluskey et al. (2018) parameterization to Figure 5.*

L458 – The comparison between AIDA and IKNA data is interesting. I have no issues with what is included in this discussion, but wonder if the authors could comment on additional impacts associated specifically with the unique particle composition. That is, once the solution droplet/particle effloresces and re-deliquesces, will it have the same ice nucleation activity as the particle that enters the AIDA chamber? Additionally,what do these results suggest for naturally occurring aerosol-cloud interactions and which (AIDA or INKA) is more representative of natural sea spray aerosol production,transport, activation and nucleation?

*The question of whether efflorescence of the particles changed their INP activity is difficult to answer and we are not sure if the algae cultures are damaged while drying. This could be interesting to investigate in future. We added the following to the manuscript in section 3.2: "Note that efflorescence might as well change the INP activity of the aerosol particles.".*
*Regarding of the question of the comparability to naturally occurring aerosol-cloud interactions, the AIDA-experiments should be more comparable to natural processes.*

Section 3.3 – See General Comment 5.
*Adapted.*

L552 – what is "mist"?
*We replaced this with "nebulised sample".*

L574 – I do not think "direct comparison" is an appropriate description for the analysis completed here because the ice nucleation ability of aerosol and bulk samples are not directly comparable. I suggest that the authors change this language to "We have normalized all of the measurements by the salt mass present in the bulk and aerosol samples to investigate the ability of ice nucleating material to transfer to the aerosol phase" or similar (see General Comment 5).

*We changed it to "In order to compare the different approaches and the ability of the ice nucleating material to transfer to the aerosol phase we have normalised all of the measurements by the salt mass present in the bulk and aerosolised samples.".*

L611 – "We also tentatively show that nebulisation enhances the ice nucleating ability of some cell cultures. We suggest that the aerosolisation process might rupture individual cells allowing ice nucleating

macro-molecules to be dispersed through the aerosol population" – Please specify that this result only has implications for laboratory studies, not reality, because the nebulizer is not a naturally occurring phenomena at the ocean surface.

*Changed to: "We suggest that the aerosolisation process using a nebuliser might rupture individual cells allowing ice nucleating macro-molecules to be dispersed through the aerosol population." We also added: "This might be unlikely to be relevant for environmental conditions.".*

L623 –This is a great discussion. While it should not and is not be expected that the authors know every paper on this topic, I do want to point out two others that are extremtely relevant to this discussion: McCluskey et al., 2018 identified two marine INP types during mesocosm experiments and also include a discussion on the timing/conditions of their emissions and a very recent paper by Wilbourn et al., 2020 describe additional phytoplankton species and may be interesting to include.

McCluskey, C. S., Hill, T. C. J., Sultana, C. M., Laskina, O., Trueblood, J., Santander,M. V., Beall, C. M., Michaud, J. M., Kreidenweis, S. M., Prather, K. A., Grassian, V.,and DeMott, P. J.: A Mesocosm Double Feature: Insights into the Chemical Makeup of Marine Ice Nucleating Particles, J. Atmos. Sci., 75, 2405-2423, 10.1175/JAS-D-17-0155.1, 2018

Wilbourn, E. K., Thornton, D. C. O., Ott, C., Graff, J., Quinn, P. K., Bates, T.S., et al.(2020). Ice Nucleation by Marine Aerosols Over the North Atlantic Ocean in Late Spring. Journal of Geophysical Research: Atmospheres, 125(4). https://doi.org/10.1029/2019JD030913

*We agree and added this two references as well.*

L629 – The heat tests would be interesting to see, especially for inferring ice nucleation material type/properties and since it is later mentioned that "The fact that marine INP are very small and heat sensitive" – is this a fact based on the data (not shown) or a hypothesis based on previous measurements?

*It is based on literature, we changed the sentence (see answer to reviewer #1).*

**References**

[revised manuscript text omitted]

---

## Author Comment (AC3) · 16 Jul 2020

The comment was uploaded in the form of a supplement:
https://www.atmos-chem-phys-discuss.net/acp-2020-246/acp-2020-246-AC3-supplement.pdf